# SCENEADAPT: SCENE-AWARE ADAPTATION OF HUMAN MOTION DIFFUSION

## ABSTRACT

Human motion is inherently diverse and semantically rich, while also shaped by the surrounding scene. However, existing motion generation approaches fail to generate diverse motion while simultaneously respecting scene constraints, since constructing large-scale datasets with both rich text-motion coverage and precise scene interactions is extremely challenging. In this work, we introduce **SceneAdapt**, a framework that injects scene awareness into text-conditioned motion models by leveraging disjoint scene–motion and text–motion datasets through two adaptation stages: inbetweening and scene-aware inbetweening. The key idea is to use motion inbetweening, learnable without text, as a proxy task to bridge two distinct datasets and thereby inject scene-awareness to text-to-motion models. In the first stage, we introduce keyframing layers that modulate motion latents for inbetweening while preserving the latent manifold. In the second stage, we add a scene-conditioning layer that injects scene geometry by adaptively querying local context through cross-attention. Experimental results show that **SceneAdapt** effectively injects scene awareness into text-to-motion models, and we further analyze the mechanisms through which this awareness emerges. Code and models will be released. Anonymous website for extensive visualizations : **link**

## 1 INTRODUCTION

Generating realistic human motion has attracted significant attention, with broad applications in virtual reality, gaming, and robotics. For practical use, motion models must satisfy two goals: achieving the *semantic richness and naturalness* of everyday actions, and ensuring *physical consistency* with the surrounding scene. Failing the former yields motions that are incoherent, while neglecting the latter produces physically implausible results, such as walking through walls. Existing approaches, however, fail to jointly ensure the semantic diversity of motion and its consistency with the scene.

On the semantic side, text-conditioned motion models (Tevet et al., 2023; Xin et al., 2023), trained on large-scale paired text–motion corpora (Punnakkal et al., 2021; Guo et al., 2022; 2025), can synthesize diverse and semantically rich motions directly from language, showing strong generalization to diverse text prompts. Yet, as these models only target text-to-motion, they remain blind to spatial constraints and cannot generate motions that interact plausibly with the environment (Fig. 1.b).

On the other hand, scene-aware motion generation aims to synthesize motions that satisfy physical constraints within the surrounding scene (e.g., collision avoidance), while remaining aligned with additional signals such as text. However, capturing motion with precise scene context typically requires professional MoCap systems, whose high cost prevents scaling to diverse scenarios. As a result, early works (Wang et al., 2022; Cao et al., 2020; Araújo et al., 2023) relied on synthetic data, and even recent motion capture datasets (Jiang et al., 2024b) remain limited to a narrow set of everyday actions (e.g., walking, sitting). Consequently, models trained on these datasets cannot generalize beyond restricted actions (Fig. 1.c).

Motivated by existing limitations, we are interested in developing a model capable of synthesizing motions that are both semantically rich and scene-aware. For example, generating "a person walking in a circle" inside a kitchen or "a man dribbling a basketball" in a bedroom requires understanding both motion semantics and spatial constraints. However, collecting large-scale text–scene–motion datasets is infeasible. We therefore formulate the task as a scene-injection problem: *How can scene awareness be incorporated into existing text-conditioned models using only scene-motion data?*

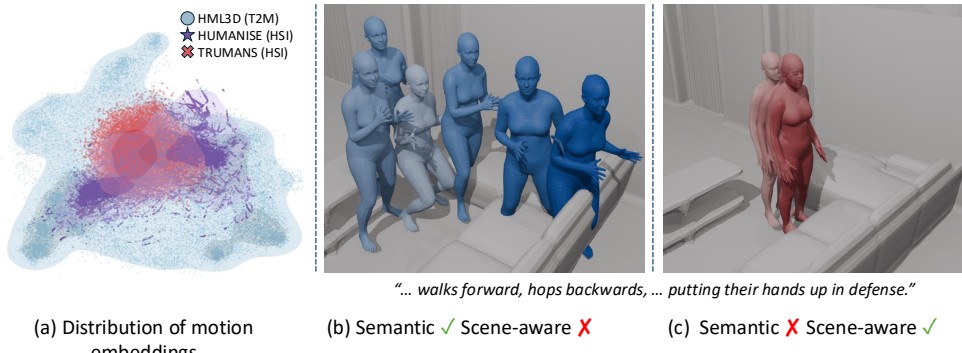

*"... walks forward, hops backwards, ... putting their hands up in defense."*

(a) Distribution of motion embeddings     (b) Semantic ✓ Scene-aware ✗     (c) Semantic ✗ Scene-aware ✓

Figure 1: **Motivation.** (a) Distribution of motion embeddings extracted with the feature extractor (Guo et al., 2022) and visualized via PCA. Scene-aware datasets (Wang et al., 2022; Jiang et al., 2024b) show narrower distributions than T2M dataset (Guo et al., 2022), indicating lower diversity and semantic coverage. (b) Models trained on T2M datasets capture diverse action semantics but lack scene awareness, penetrating the obstacles. (c) Models trained on scene-aware datasets satisfy scene constraints but fail to follow text conditions.

In this paper, we introduce **SceneAdapt**, a two stage adaptation framework that injects scene awareness into a pretrained motion diffusion model (MDM), using only existing text-motion and scene-motion datasets. Our key insight is to leverage motion inbetweening, which can be learned without text, as a proxy task to inject scene awareness and enable scene-aware text-conditioned generation. To be specific, we first adapt MDM to motion inbetweening, and then further adapt it for scene-aware inbetweening (Hwang et al., 2025) using only scene–motion pairs. Since the model already learns inbetweening in the first stage, the second stage focuses exclusively on leveraging scene data to achieve scene-consistent inbetweening, thereby injecting scene awareness into the model.

To adapt text-conditioned models for inbetweening, we design a Context-aware Keyframing (CaKey) layer that selectively modulates keyframe latents, enabling accurate inbetweening while preserving the pretrained latent manifold. In the second stage, we freeze the CaKey layer and insert Scene Conditioning layers that use cross attention to enable scene-awareness. Whereas prior works (Jiang et al., 2024b) use global features, we utilize patch-wise features, allowing frame-wise latent to focus on different places in the scene. Through these adaptations, the model can generate motions that are both faithful to text prompts and consistent with the surrounding scene.

Extensive experiments demonstrate that **SceneAdapt** genuinely exploits scene information, leading to motions that are both semantically rich and scene-aware. Furthermore, we show that proposed components at each stages lead to significant performance gain, validating that our overall pipeline is effective. We further analyze how scene awareness is injected into the model, providing new insights into the mechanisms through which text-conditioned motion generation benefits from scene information.

The main contributions of this work are: (1) We propose **SceneAdapt**, a two-stage adaptation framework that injects scene awareness into a pretrained motion diffusion model using only text-motion and scene-motion datasets. (2) We design **Context-aware Keyframing (CaKey) layer**, which modulates only keyframe latents to enable faithful motion inbetweening without distorting the latent manifold. (3) We introduce a **Scene-conditioning layer** that leverages cross-attention between frame-wise motion latents and voxel patch features. (4) Extensive experiments show that **SceneAdapt** outperforms scratch-trained baselines, improves scene awareness in text-to-motion generation, and provides insights into how scene information is integrated into generative models.

## 2 RELATED WORK

### 2.1 SYNTHESIZING HUMAN MOTION

**Text-to-motion (T2M) synthesis.** Given a text description, this field aims to generate corresponding natural and diverse motions. Early works employed models such as RNNs or Transformers (Guo

et al., 2020; Petrovich et al., 2022; Zhang et al., 2023a; Siyao et al., 2022; Zhang et al., 2023b), and focused on alignment between motion and language latent spaces (Ahuja & Morency, 2019; Tevet et al., 2022). Recently, Tevet et al. (2023) introduced the Motion Diffusion Model (MDM), a text-conditioned motion generator trained on large-scale text–motion datasets such as Plappert et al. (2016) and Guo et al. (2022), demonstrating strong generative performance. Subsequent works (Zhong et al., 2023; Xin et al., 2023; Dai et al., 2024; Pinyoanuntapong et al., 2024b;a; Barquero et al., 2024; Guo et al., 2024a; Cho et al., 2025) have further improved generation quality, efficiency, and semantic alignment. However, as these datasets lack scene context, the resulting models remain inherently unaware of their surroundings.

**Scene-aware T2M synthesis.** Scene-aware text-to-motion aims to generate motions that are not only natural and faithful to the textual description but also physically consistent with a 3D scene. However, obtaining real motion data that is accurately aligned with surrounding scene geometry remains extremely challenging. To address this limitation, several recent works (Black et al., 2023; Araújo et al., 2023; Wang et al., 2022; Yi et al., 2024; Cen et al., 2024) construct synthetic scene–motion datasets as a scalable alternative to expensive real-world capture. For instance, HU-MANISE (Wang et al., 2022) introduced a large-scale synthetic dataset by aligning the scanned indoor scenes with captured motion sequences. Although such datasets enable scalable training like (Wang et al., 2024a), they fall short in capturing the realism of actual human–scene interactions. Recent works (Jiang et al., 2024b;a; Zhang et al., 2024; Araújo et al., 2023; Cong et al., 2024) have proposed real-world MoCap datasets captured with professional apparatus. However, these datasets remain impractical due to their limited action diversity (e.g., walking, sitting down, picking, standing up) and are difficult to scale up because of their high cost. To avoid the reliance on datasets, some studies (Li & Dai, 2023; Li et al., 2025) leverage pretrained image or video diffusion models for zero-shot motion generation, but struggle to generate realistic motions. Instead of collecting new datasets or indirect solutions, we leverage existing motion corpora and introduce a novel adaptation strategy that enables the model to produce semantically rich motions while simultaneously adhering to the geometry and constraints of the given scene.

**Spatially controlled T2M synthesis.** Several studies (Karunratanakul et al., 2023; Zhao et al., 2025; Ron et al., 2025) have focused on spatial control by propagating gradients from external conditions, such as pelvis trajectories, 2D obstacles or even objects, into the diffusion noise. However, these methods require extra computation during optimization, leading to slow synthesis. Moreover, they often fail to reflect the text descriptions, as satisfying scene constraints takes priority. In contrast, we present a feed-forward approach that generates motions that are both scene-aware and faithful to text conditions.

## 2.2 ADAPTATION OF DIFFUSION MODELS

Diffusion models pretrained on large-scale datasets (Rombach et al., 2022; Peebles & Xie, 2023) demonstrate impressive generative ability, but often require adaptation to new conditions or domains. One representative method is ControlNet (Zhang et al., 2023c), which augments a frozen network with a trainable copy, enabling generation guided by various signals such as pose, edge or depth maps. Another widely used strategy is LoRA (Hu et al., 2021), which adapts pretrained diffusion models to novel domains in a parameter-efficient manner (Guo et al., 2024b; Shi et al., 2024). Recent efforts introduce auxiliary modules to incorporate additional control signals such as camera parameters (He et al., 2025; Wang et al., 2024b) or user action controls (Yu et al., 2025). Among these, (Yu et al., 2025) introduces multi-phase adaptation pipeline, which motivates our strategy. While (Yu et al., 2025) adapts a video diffusion model to respond to interactive controls like keyboard inputs, our method instead equips text-conditioned motion diffusion model with 3D scene awareness.

## 3 METHOD

The overall pipeline of SceneAdapt is illustrated in Fig. 2. We first adapt MDM for *motion in-betweening* (§ 3.2) using our novel CaKey layer, which generates natural motions consistent with input keyframes. Next, we freeze the CaKey layers and insert scene-conditioning layers (SceneCo) to learn scene-aware inbetweening (§ 3.3). At inference, we use the trained adapters to perform scene-aware text-to-motion generation (§ 3.4). For implementation details, see Appendix C .

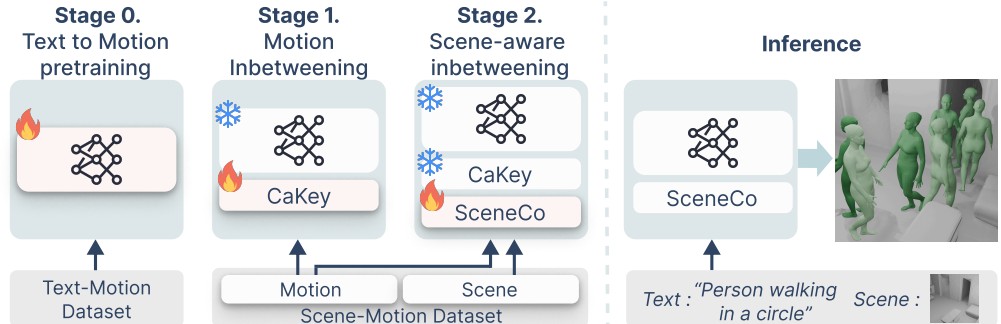

Figure 2: **Overview.** Starting from a pretrained text-to-motion model (**Stage 0**), we first insert CaKey layers and train them with a motion inbetweening objective (**Stage 1**), which only requires motion sequences. We then add scene-conditioning layers (denoted SceneCo) and train them with a scene-aware inbetweening objective (**Stage 2**), using scene-motion pairs. During inference, we only use the base model and SceneCo layers for scene-aware text-to-motion generation.

### 3.1 PRELIMINARIES

**Problem Formulation.** We define a 3D scene as $\mathcal{S}$, a text prompt as $\mathcal{T}$, and a keyframe mask $m^{1:N} = \{m^i\}_{i=1}^N$ with $m^i \in \{0, 1\}$, where $m^n = 1$ indicates that the $n^{th}$ frame is a keyframe. Our goal is to generate a natural motion sequence $x^{1:N} = \{x^i\}_{i=1}^N$, where $x^i \in \mathbb{R}^D$, conditioned on different forms of context: (i) *motion inbetweening*, which models $p(x^{1:N} \mid m^{1:N})$; (ii) *scene-aware inbetweening*, which models $p(x^{1:N} \mid m^{1:N}, \mathcal{S})$; (iii) *scene-aware text-conditioned generation*, which models $p(x^{1:N} \mid \mathcal{S}, \mathcal{T})$.

**Motion Representation.** We adopt the HML3D (Guo et al., 2022) representation, where each pose $x^i$ is a 263-dimensional vector. Following (Cohan et al., 2024), we convert the relative root orientation and the relative $x, z$ positions into their global counterparts, which allows us to adapt MDM for motion inbetweening.

**Motion Diffusion Model.** We adopt MDM (Tevet et al., 2023) as our baseline model, following prior works (Sawdayee et al., 2025; Xie et al., 2024; Karunratanakul et al., 2024). MDM models text-conditioned motion generation within the DDPM framework (Ho et al., 2020), which consists of a forward and backward diffusion processes. The forward diffusion is formulated as a Markov noising process that produces a sequence $\{x_t\}_{t=0}^T$, where $x_0$ is the clean data and $t$ is the diffusion timestep. Each step is defined as $q(x_t \mid x_{t-1}) = \mathcal{N}(x_t; \sqrt{1 - \beta_t}\, x_{t-1}, \beta_t \mathbf{I})$, with $\{\beta_t\}_{t=1}^T$ denoting the variance schedule. During the backward pass, instead of predicting the noise $\epsilon$, the denoising network is parameterized to directly predict the clean motion $\hat{x}_0 = \mathcal{D}_\theta(x_t, t, \mathcal{T})$. The training objective is the simplified $L_2$ reconstruction loss,

$$\mathcal{L}_{\text{t2m}} = \mathbb{E}_{x_0 \sim q(x_0 \mid \mathcal{T}),\, t \sim [1, T]} \left[ \left\| x_0 - \mathcal{D}_\theta(x_t, t, \mathcal{T}) \right\|_2^2 \right]. \tag{1}$$

with additional geometric losses applied in the raw motion space.

### 3.2 STAGE 1: ADAPTATION FOR INBETWEENING

Prior approaches have explored inbetweening either by imputing keyframes at inference time (Tevet et al., 2023) or by training specialized models from scratch (Cohan et al., 2024; Hwang et al., 2025). However, the adaptation of text-conditioned motion generation models to the inbetweening setting remains unexplored. A well-adapted inbetweening model should not only achieve high keyframe alignment, but also preserve the naturalness and text-adherence capability of the original model.

**Adaptation Layers.** To achieve these properties, we introduce the Context-aware Keyframing (CaKey) layer, which applies affine modulation to the MDM latents based on the given keyframes. Formally, CaKey employs two learnable MLP-based networks, $f_\theta$ and $h_\phi$. These networks take as input the keyframe mask $m$, ground-truth motion $x$, the diffusion timestep $t$, and the current self-attention activation $a$, and output the scale $\gamma$ and shift $\beta$ parameters:

$$\gamma = f_\theta(x, t, a), \quad \beta = h_\phi(x, t, a) \tag{2}$$

Our modulation process is described as

$$\hat{a} = \gamma \odot a + \beta, \tag{3}$$

$$\text{CaKey}(a, m, x, t) = (1 - m) \odot a + m \odot \hat{a}, \tag{4}$$

CaKey introduces two key modifications over standard FiLM-style modulation: (1) *Context-awareness*. The modulation parameters are estimated not only from the keyframe signal but also from the diffusion timestep along with the latent representation being modulated, enabling the modulation to be aware of what it is modulating, and thereby improving alignment with input keyframes. (2) *Sparse modulation*. Identity is preserved on non-keyframe indices while modulation is applied only on the keyframe indices, ensuring that only the keyframe latents are modulated.

**Training.** We freeze the base MDM parameters and optimize only the CaKey layers under the motion inbetweening objective. The loss follows the diffusion formulation in Eq. 1, with two modifications: (i) the text input is replaced by the null embedding $\varnothing_{text}$, and (ii) conditioning is augmented with a keyframe mask $m^{1:N}$. The mask is

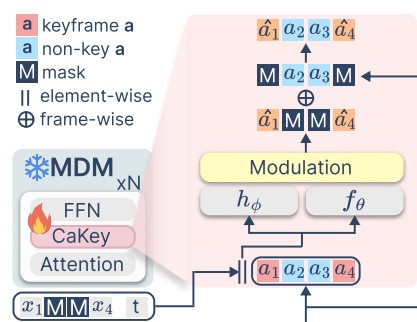

Figure 3: **CaKey Layer.** Text-to-motion models adapted with CaKey exceed the performance of inbetweening models trained from scratch.

sampled randomly with a fixed stride $s_k$, while the first and last frames are always designated as keyframes ($m^0 = m^N = 1$). In all experiments on stage 1, we set $s_k = 20$ which corresponds to one keyframe per second.

### 3.3 STAGE 2: ADAPTATION FOR SCENE-AWARE INBETWEENING

Building on the inbetweening adaptation, we introduce additional layers for scene conditioning (SceneCo) and train them while keeping the rest of the model frozen. It is important to emphasize that the learned inbetweening capability allows scene-aware learning to become the primary objective in minimizing the training loss. This design thus encourages the new parameters to focus solely on leveraging scene information, thereby injecting scene-awareness into the model.

**Scene Representation.** Previous approaches encode $\mathcal{V}$ into a single *global vector* via the class embedding of a Voxel ViT (Jiang et al., 2024b;a), conditioning all frames on the same vector (Hwang et al., 2025). However, such global features overlook the fact that joint positions evolve over time, and thus different frames interact with different local neighborhoods of the scene. To capture this spatio-temporal variation, we propose to use patch embeddings from a voxel ViT and then enable interactions between motions and these patches. At first, we voxelize the scene $\mathcal{S}$ into a binary occupancy grid $\mathcal{V} = \text{voxelize}(\mathcal{S}) \in \{0, 1\}^{d_x \times d_y \times d_z}$, where 1 denotes an occupied cell and 0 a free one. Then, we obtain patch embeddings from ViT: $s = \text{ViT}(\mathcal{V}) \in \mathbb{R}^{P \times d_s}$, where $P$ is the number of spatial patches and $d_s$ the embedding dimension. These patch-level tokens enable each frame to dynamically attend to its spatially relevant context, rather than relying on a static global vector.

**Adaptation Layers.** To bridge two modalities, spatial embeddings and temporal motion latents, we employ SceneCo layers. These layers are cross-attention layers, where motion latents query voxel patches so that each frame can selectively attend to its relevant local context. Formally, let $h = \{h^i\}_{i=1}^{N+1} \in \mathbb{R}^{(N+1) \times d}$ denote the latent sequence, where $h^1$ corresponds to the text token and $\{h^i\}_{i=2}^{N+1}$ to motion frames. Let $s = \{s^j\}_{j=1}^{p_n} \in \mathbb{R}^{p_n \times s_{dim}}$ be the patch embeddings of the voxelized scene. Cross-attention is then defined as

$$h_{out} = \text{ATT}(hW_Q, sW_K, sW_v)$$

To ensure that scene information is used only where necessary, we mask activations as follows: (i) the text token $h^1$ and (ii) padded frames, leaving scene conditioning active only for motion latents that require scene awareness.

**Training.** We keep the MDM and CaKey layers frozen, and train the additional cross-attention layers along with our voxel ViT on the motion inbetweening objective using 3D scene as inputs. A

key challenge at this stage is that, unlike the previous stage where only keyframes are modulated, the cross-attention layers broadly affect the motion latent space, leading to a decline in the model's original text-to-motion performance. To mitigate this issue, we utilize the text-motion paired dataset used during pretraining for prior preservation (Ruiz et al., 2023; Sawdayee et al., 2025), by adding Eq. 1. As text-motion paired dataset do not provide 3D scenes, we introduce a learnable null embedding $\varnothing_{scene}$ for the prior loss, while dropping 10% of text inputs for classifier-free guidance. We also apply $\varnothing_{scene}$ for 10% of the scene features in the scene-motion pairs.

### 3.4 TEXT TO SCENE-AWARE MOTION GENERATION

With both CaKey layers and SceneCo layers trained, we perform scene-aware text-conditioned motion generation by conditioning the final model only on text and scene inputs, while using an all-zero keyframe mask ($m^{1:N} = 0$), indicating that no keyframes are provided.

**Sampling.** We introduce two classifier-free guidance scales: $w_t$ for text guidance and $w_s$ for scene guidance. These scales control the trade-off between semantic alignment with the text and physical consistency with the scene during motion generation. Formally,

$$\hat{x}_0 = \mathcal{D}_\theta(x_t, t, \varnothing_{text}, \varnothing_{scene}) + w_t\big(\mathcal{D}_\theta(x_t, t, \mathcal{T}, \varnothing_{scene}) - \mathcal{D}_\theta(x_t, t, \varnothing_{text}, \varnothing_{scene})\big)$$
$$+ w_s\big(\mathcal{D}_\theta(x_t, t, \varnothing_{text}, \mathcal{S}) - \mathcal{D}_\theta(x_t, t, \varnothing_{text}, \varnothing_{scene})\big). \tag{5}$$

**Goal pose conditioning.** Our objective is to generate motions that are semantically rich and scene-aware (e.g., avoiding penetration with the environment). However, because our approach does not model scene–semantic relationships (e.g., "walk to the refrigerator"), it cannot directly produce functional behaviors that require semantic understanding of the scene. Nevertheless, since our adaptation method is based on sparse keyframing, we can achieve goal-directed, scene-aware motion by additionally conditioning the model on goal poses. Under this setting, the model is able to sit on chairs and reach toward objects across diverse scenes when provided with a goal pose, the scene, and a text description. The results are shown in our **project website** and Figure 5. Leveraging scene-aware pose generation methods to further guide SceneAdapt toward functional, goal-directed motion represents a promising direction for future work.

## 4 EXPERIMENTS

We first evaluate SceneAdapt on scene-aware text-conditioned motion generation (§ 4.1), then assess the effectiveness of CaKey for motion inbetweening, and further examine how incorporating scene-conditioning layers injects scene-awareness (§ 4.2). Finally, we conduct a component-wise ablation study to validate the contribution of each design choice (§ 4.3).

**Dataset.** The baseline MDM model is trained on the text–motion paired HumanML3D dataset (Guo et al., 2022). For adaptation, we additionally use the scene–motion paired TRUMANS dataset (Jiang et al., 2024b), the largest high-quality mocap dataset with precise alignment to scene geometry. While HumanML3D is represented using the skeleton from SMPL-H (Romero et al., 2017), TRUMANS is provided in SMPL-X (Pavlakos et al., 2019). To ensure compatibility, we fit SMPL-H meshes to TRUMANS motions and follow the preprocessing pipeline of (Guo et al., 2022). TRUMANS sequences are relatively slow and long, recorded at 30 FPS. We downsample them by a factor of 2 (15 FPS) and segment them into 196 frame clips. Although HumanML3D is at 20 FPS, the slower dynamics of TRUMANS make the downsampled sequences match the speed of HumanML3D. For evaluation, since no text–scene–motion paired dataset with diverse textual descriptions exists, we augment the HumanML3D test set by randomly matching each motion-text pair with a sampled trajectory position and rotation from TRUMANS motion sequences, which serve as the first frame's global position and rotation. This yields pseudo text–scene–motion pairs that enable us to evaluate text-to-motion generation using established metrics as well as our scene-aware metrics. Details of this evaluation set construction are provided in Appendix A.

**Evaluation Metrics.** We compute Frechet Inception Distance (FID) to measure the overall diversity and naturalness of the generated motions and R-Precision (RP) (Guo et al., 2022) to evaluate the text-adherence to the given prompt. For inbetweening, we quantify the mean joint position error (MJPE) for both the full sequence and the keyframes to measure keyframe alignment. We further

| Method | Optimization | Dataset | R-P (top 3)↑ | FID↓ | CFR↓ | MMP↓ | JCR↓ | Inf. Time (s)↓ |
|---|---|---|---|---|---|---|---|---|
| MDM | ✗ | HM | 0.798 | 0.479 | 0.316 | 0.319 | 0.344 | 0.52 |
| DNO | ✓ | HM | 0.128 | 32.22 | 0.001 | 0.002 | 0.002 | 332.96 |
| DARTControl | ✓ | HM | 0.056 | 53.29 | 0.010 | 0.007 | 0.010 | 362.90 |
| AffordMotion | ✗ | HU | 0.140 | 21.59 | 0.257 | 0.059 | 0.097 | 50.72 |
| AffordMotion | ✗ | HM+HU | 0.305 | 6.320 | 0.429 | 0.254 | 0.321 | 51.28 |
| Ours ($w_s = 0.3$) | ✗ | HM+TR | 0.792 | 0.497 | 0.256 | 0.208 | 0.246 | 1.69 |
| Ours ($w_s = 0.0$) | ✗ | HM+TR | 0.803 | 0.312 | 0.298 | 0.273 | 0.299 | |
| Ours ($w_s = 0.5$) | ✗ | HM+TR | 0.750 | 1.420 | 0.220 | 0.160 | 0.199 | |
| Ours ($w_s = 1.0$) | ✗ | HM+TR | 0.588 | 7.389 | 0.136 | 0.076 | 0.101 | |
| Ours ($w_s = 2.0$) | ✗ | HM+TR | 0.365 | 18.88 | 0.072 | 0.035 | 0.045 | |

Table 1: **Scene-aware text-driven generation results** on our evaluation set. "Dataset" shows the primary training dataset (HM = HumanML3D, HU = HUMANISE, TR = TRUMANS), and "Inf. Time" reports the average inference time per sample in RTX A5000.

report foot skating (Karunratanakul et al., 2023) and skating ratio (Zhang & Tang, 2022) to quantify sliding artifacts. Motivated by prior scene-aware works (Zhang et al., 2020; Hwang et al., 2025; Wang et al., 2022), we holistically assess geometry compliance using 3 metrics. Collision-frame ratio (**CFR**) measures *how often* violations occur: the fraction of frames with any penetration. mean max penetration (**MMP**) measures *how severe* a violation is when it happens: the average per-frame deepest penetration (m) over colliding frames. Joint-collision ratio (**JCR**) measures *how widespread* a violation is: the mean fraction of joints penetrating, computed *only over colliding frames* (pure extent), thus decoupled from CFR's frequency. We define penetration using signed distance fields (SDFs) with a 2 cm tolerance: letting $d_{t,v}$ be the signed distance of joint $v$ at frame $t$ (negative inside), a joint is counted as colliding iff $d_{t,v} < -\delta$ with $\delta = 2$ cm.

**Baselines.** For scene-aware text-to-motion generation, we compare SceneAdapt against state-of-the-art optimization-based methods DNO (Karunratanakul et al., 2024) and DART (Zhao et al., 2025), as well as the feed-forward method AffordMotion (Wang et al., 2024a). For motion inbetweening, we benchmark against imputation-based sampling (Tevet et al., 2023), LoRA (Hu et al., 2021), and CondMDI (Cohan et al., 2024), a model specifically designed for inbetweening.

### 4.1 SCENE-AWARE TEXT CONDITIONED MOTION GENERATION

**Quantitative results.** As shown in Tab. 1, compared to MDM, our adaptation improves its scene-awareness without sacrificing its text-to-motion capabilities. Compared to AffordMotion, our method achieves superior performance in both text-to-motion alignment and scene-awareness (see Ours $w_s = 0.5$), showing that training solely on high-quality scene–motion pairs can outperform models trained with synthetic text–scene–motion triplets of limited semantic coverage. Furthermore, while optimization-based methods often fail to preserve the original model's generative capabilities, SceneAdapt not only retains them but even surpasses the baseline model, despite being adapted exclusively for inbetweening (see Ours $w_s = 0.0$). Although optimization-based approaches achieve nearly perfect scene-awareness by directly optimizing motions with respect to the evaluation metrics, their inference time is roughly $200\times$ slower than ours. Overall, SceneAdapt combines high scene-awareness with strong text alignment and naturalness, while remaining orders of magnitude faster than optimization-based baselines, making it a practical solution for scalable scene-aware motion generation.

**Qualitative results.** As shown in Fig. 4, AffordMotion suffers from scene penetration or weak adherence to text prompts, reflecting limitations of the HUMANISE dataset, which contains only synthetic scene–motion interactions with limited diversity. DNO achieves strong scene-awareness but sacrifices text alignment during the optimization process. In contrast, our method equips MDM with scene-awareness, substantially reducing scene penetration while preserving text fidelity. Moreover, we show that conditioning on an additional goal pose enables our model to generate motions that interact with the scene without penetrating it, while still following the text prompt.

### 4.2 MOTION INBETWEENING

**Quantitative results.** We report quantitative comparisons between our first-stage model and other baselines, as summarized in Table 2. Simply applying imputation at inference yields suboptimal

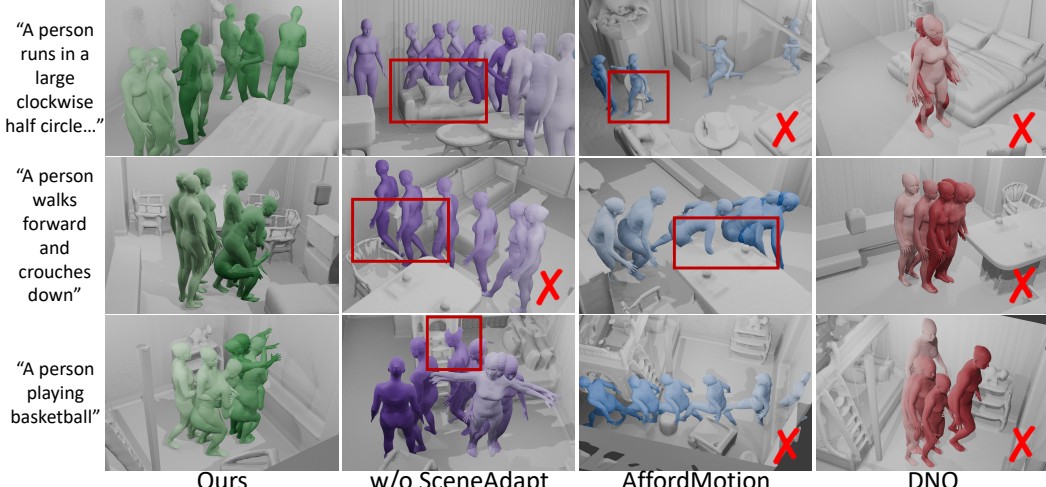

Figure 4: **Qualitative results** on our evaluation set, where red boxes mark collisions and red X's mark semantic errors; unlike AffordMotion (scene penetration) and DNO (weak text alignment), our method improves MDM by enhancing scene-awareness while preserving text fidelity.

| Method | R Precision (top 3)↑ | FID↓ | MJPE(Key)↓ | MJPE(All)↓ | Foot skating↓ | Skating ratio↓ |
|---|---|---|---|---|---|---|
| GT | 0.7980 | 0.002 | 0 | 0 | - | - |
| MDM (imputation) | 0.6144 | 7.258 | 0 | 0.7647 | 0.1012 | 0.3971 |
| MDM + LoRA | 0.7214 | 0.074 | 0.0625 | 0.1120 | 0.0418 | 0.0625 |
| CondMDI | 0.6767 | 0.356 | 0.2804 | 0.2957 | 0.1067 | 0.1074 |
| Ours | 0.7242 | 0.036 | 0.0018 | 0.0550 | 0.0479 | 0.0623 |
| w/o time embedding | 0.7197 | 0.0369 | 0.0017 | 0.0536 | 0.0481 | 0.0638 |
| w/o adaptivity | 0.7220 | 0.0548 | 0.0038 | 0.1028 | 0.0527 | 0.0638 |
| w/o sparse modulation | 0.2015 | 17.442 | 0.0007 | 0.650 | 0.0560 | 0.0626 |

Table 2: **Motion inbetweening results** on the HML3D test set. Our CaKey design outperforms imputation sampling, LoRA, and CondMDI, highlighting the importance of context-aware modulation.

results, indicating that the generative prior of MDM cannot cover the sparsity of keyframes for inbetweening tasks, thus requiring further adaptation. While LoRA (Hu et al., 2021) is effective, it still underperforms due to the lack of modules specifically designed for inbetweening. CondMDI (Cohan et al., 2024), trained from scratch for inbetweening, also yields inferior results compared to ours. This result highlights the effectiveness of our CaKey design, whereas CondMDI merely concatenates keyframe masks with input motions, the CaKey layer leverages richer signals to modulate only the keyframe latents. Furthermore, we validate each design choice within the CaKey layer, with results showing that every component contributes critically to its overall effectiveness. Extensive ablations on our design on CaKey can be found in Appendix B.

**Scene-awareness during Inbetweening.** While the keyframe stride $s_k$ is fixed at 20 in stage 1, we vary $s_k$ when training the scene-conditioning layer, and evaluate with $s_k$ used in stage 2 to examine improvements in scene-awareness. As shown in Tab. 3, using the same $s_k$ as stage 1 yields similar collision rates, since the model is already adapted specifically for motion inbetweening, leaving little room to improve. However, increasing $s_k$ encourages the scene-conditioning layer to more effectively exploit scene information, resulting in larger gains under sparser keyframes.

| $s_k$ | Stage. | CFR↓ | MMP↓ |
|---|---|---|---|
| 20 | stage 1 | 0.021 | 0.011 |
| | stage 2 | 0.022 (−5%) | 0.010 (+9%) |
| 40 | stage 1 | 0.030 | 0.016 |
| | stage 2 | 0.030 (0%) | 0.014 (+13%) |
| 60 | stage1 | 0.037 | 0.020 |
| | stage 2 | 0.033 (+11%) | 0.015 (+25%) |
| 80 | stage1 | 0.054 | 0.028 |
| | stage 2 | 0.040 (+26%) | 0.019 (+32%) |

Table 3: **Scene-awareness results** on the TRUMANS test set for inbetweening.

"A person sits down on the chair in front of the table."

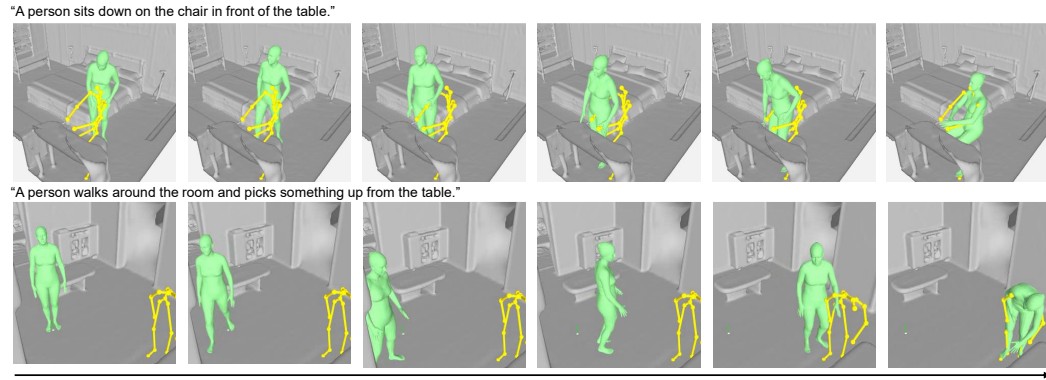

"A person walks around the room and picks something up from the table."

time

Figure 5: **Goal pose conditioned scene-aware text-to-motion generation.** Interpreting the goal pose as an extremely sparse keyframe, SceneAdapt produces scene-consistent motion conditioned on text, scene, and goal pose. Goal poses are visualized in yellow.

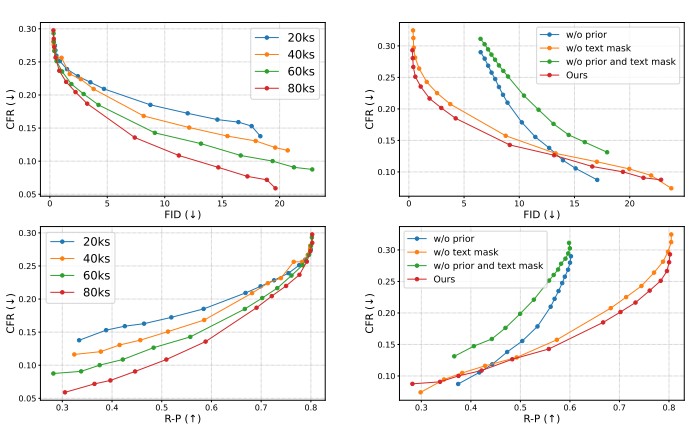

| Emb. ($w_s$) | R-P↑ | CFR↓ | MMP↓ |
|---|---|---|---|
| Class (0) | 0.798 | 0.314 | 0.308 |
| Patch (0) | 0.802 | 0.293 | 0.258 |
| Class (0.1) | 0.799 | 0.306 | 0.299 |
| Patch (0.1) | 0.800 | 0.281 | 0.242 |
| Class (0.3) | 0.796 | 0.290 | 0.281 |
| Patch (0.3) | 0.783 | 0.251 | 0.200 |
| Class (0.4) | 0.794 | 0.285 | 0.272 |
| Patch (0.4) | 0.761 | 0.236 | 0.178 |

(c) **Effect of Scene Representation.** Voxel patch features outperform class embeddings.

| Strategy | FID↓ | R-P↑ |
|---|---|---|
| w/o inbetween | 7.08 | 0.598 |
| Ours | 0.497 | 0.791 |

(a) **Effect of Keyframe Strides (KS).** Sparser keyframes on stage 2 force the model to better exploit the scene, leading to an increase in scene-awareness.

(b) **Effect of Prior Preserving Designs.** The prior loss and the text mask for cross attention on stage 2 help the adaptation preserve the original t2m capabilities.

(d) **Effect of Inbetweening Adaptation.** Inbetweening is crucial for scene-motion only adaptation.

Figure 6: **Ablation Studies on Scene-aware Inbetweening.** Each dot indicates a different scene CFG weight ($w_s$) ranging from 0.001 to 2.5. Text CFG weights are fixed to 2.5.

## 4.3 ANALYSIS

**Keyframe Stride at Stage 2.** We ablate how varying $s_k$ in stage 2 impacts scene-aware text-to-motion performance. As shown in Fig. 6a, sparser keyframes consistently improve performance, aligning with the results from scene-aware inbetweening (Tab. 3). This indicates that the model leverages scene awareness acquired in stage 2 and transfers it to text-conditioned motion generation.

**Prior Preservation.** We analyze how the prior loss and text mask preserve the original capabilities of MDM by evaluating on the scene-aware text-to-motion task. As shown in Fig. 6b, incorporating the prior loss significantly improves T2M performance, while using the text mask during adaptation provides additional gains. Furthermore, Tab. 6d shows that when adapting MDM without text, inbetweening is crucial for preserving its original capabilities.

**Scene-conditioning Layer.** Using patch embeddings instead of class embeddings proves more effective for injecting scene awareness, as shown in Tab. 6c. To understand why, we analyze attention weight maps of our scene-conditioning layer in Fig. 7. Occupied regions near the human receive high attention values, while empty regions nearby receive relatively low values. Moreover, attention weights vary dynamically along the human's trajectory. These patterns suggest that motion latents interact with patch embeddings in a spatially adaptive manner through the cross-attention layers.

**User study.** We conduct user study with 28 participants to provide further analysis. Participants are

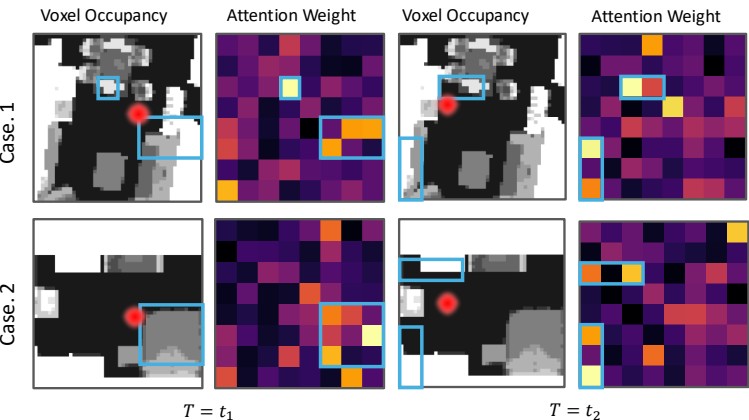

Figure 7: **Visualization of the cross-attention weight map between the motion latent at a specific timestep and the patch-wise scene embeddings.** The red point indicates the human location.

| Method | Text-adherence ↑ | Collision-avoidance ↑ |
|---|---|---|
| MDM | 0.147 | 0.040 |
| AffordMotion (All) | 0.024 | 0.016 |
| DNO | 0.040 | 0.119 |
| **Ours** | **0.790** | **0.825** |

Table 4: **User study results on scene-aware motion generation.** Numbers denote the preference rate (fraction of trials in which each method was selected as the best motion; higher is better) for text adherence and collision avoidance.

requested to answer (1) which motion best matches the text description (text adherence), (2) which motion best avoids collisions with the scene and obstacles (collision avoidance). Motions from ours and baseline models are shown in random order during the study.

## 5 LIMITATIONS AND CONCLUSIONS

A key limitation is the absence of a public dataset that simultaneously supports evaluating both motion semantics (e.g., FID, R-P) and scene-geometry awareness (e.g., CFR, MMP, JCR) using ground-truth data. To approximate such an evaluation setting, we construct our own paired dataset by matching ground-truth text–motion pairs with scenes. However, this pairing pipeline can introduce bias toward certain prompts. Although we apply SDF-based filtering to remove clearly implausible pairings (e.g., "run forward" starting directly in front of a wall), some prompts naturally benefit from certain scene configurations. For example, "a person running" paired with a large open room will exhibit less penetration than if placed in a small, cluttered space. Still, our constructed evaluation set provides a reasonable proxy for assessing both motion semantics and scene-awareness. Nevertheless, building a large-scale ground truth text-scene-motion pairs large enough to evaluate model-based metrics such as FID would be an important direction for future work.

We introduced **SceneAdapt**, a two-stage adaptation framework that injects scene awareness into pretrained text-to-motion diffusion models. Our key idea is to use motion inbetweening as a bridge to leverage both text–motion and scene–motion datasets, avoiding the need for costly large-scale text–scene–motion collections. In the first stage, the model is adapted for motion inbetweening through our Context-aware Keyframing (CaKey) layer, while in the second stage, scene awareness is incorporated via scene-conditioning layers. Together, these adaptations enable the generation of motions that are both semantically rich and physically consistent with surrounding scenes. Extensive experiments confirm the effectiveness of each stage and validate the overall strength of the framework.

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

APPENDIX

We refer the reader to the accompanying videos for extensive qualitative results on scene-aware text-conditioned motion generation.

## A    EVALUATION SET CONSTRUCTION

We construct the evaluation set consisting of text–scene–motion pairs as follows. We first extract the 3D coordinates of the pelvis joint and their corresponding SDF values from each frame of the TRUMANS dataset, where the SDF fields are preprocessed from the provided scene meshes (Jiang et al., 2024b). Frames with root heights indicating sitting or lying are discarded, as such low starting positions violate the canonicalization of motion data. Among the remaining frames, we remove those with low SDF values since they correspond to humans standing too close to surrounding objects, which can lead to implausible synthesis. For example, it is unnatural to generate motion when "a person runs forward" is given as text condition but the starting point is already immediately in front of the wall. We then sort the valid frames by SDF and keep the top $10\%$, using the pelvis positions of these frames as initial points. To provide motion and text annotations, we randomly sample motions from HumanML3D while excluding climbing or stair-related actions, which do not exist in TRUMANS. Following this procedure, we obtain 3K text–scene–motion pairs for evaluation.

## B    DETAILED ABLATIONS

| Sparse Mod. | Adaptive | Time emb. | Modulator | FID↓ | MJPE (Key)↓ | MJPE(All)↓ |
|:---:|:---:|:---:|:---:|:---:|:---:|:---:|
| ✓ | ✓ | ✓ | MLP | 0.0356 | 0.0018 | 0.055 |
|  | ✓ | ✓ | MLP | 17.442 | 0.0007 | 0.650 |
| ✓ | ✓ |  | MLP | 0.0369 | 0.0017 | 0.0536 |
| ✓ |  | ✓ | MLP | 0.0548 | 0.0038 | 0.1028 |
| ✓ | ✓ | ✓ | Linear | 0.0485 | 0.0027 | 0.0764 |
| ✓ |  | ✓ | Linear | 0.0924 | 0.0051 | 0.1308 |
| ✓ | ✓ |  | Linear | 0.0485 | 0.0027 | 0.0764 |
| ✓ |  |  | Linear | 0.0849 | 0.0044 | 0.1173 |

Table 5: **Ablation study on motion inbetweening designs.** Sparse Mod. indicates whether sparse modulation is used. Adaptive denotes whether the source latent is provided as input to the modulator. Time emb. specifies whether time embedding is provided as input to the modulator. Modulator describes how $f_\theta$ and $h_\phi$ are modeled.

**Ablations on Cakey components.** As reported at Table 5, We ablate key components of CaKey layer introduced in 3.2. One crucial element is the sparse modulation, which focuses on keyframe poses while preserving the non-keyframe latents. Replacing it with global modulation (second row) results in a significant performance drop, validating its effectiveness. As shown in the third and fourth rows, leveraging contextual signals such as source latent motion or timestep embeddings is also critical. Finally, the network design of the modulator is important for fully utilizing these contexts, as models with MLPs consistently outperform those with linear layers.

## C    IMPLEMENTATION DETAILS

**MDM Pretraining.** As shown in Cohan et al. (2024), using motion representations with global root information can lead to severe foot skating results, which can be alleviated by adopting a U-Net architecture (Karunratanakul et al., 2023) instead of the transformer architecture originally used in MDM. In our experiments, we found that introducing additional global position and velocity losses significantly improves motion naturalness, achieving the same performance to the original motion

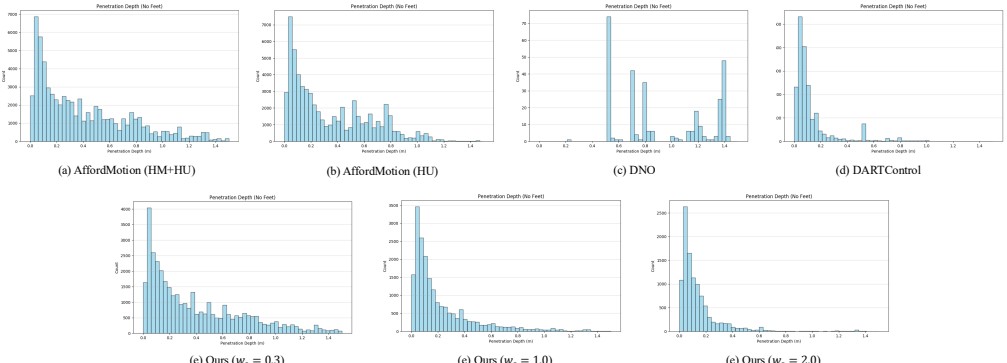

Figure 8: **Distribution of Penetration-Depth.** This figure shows the distribution of the per-frame maximum penetration depth over colliding frames (unit: m). Note that the total count differs across histograms because frames without any penetration are excluded from this plot. Please zoom in to see the details.

| Method | ZR (%) ↑ | P50 ↓ | P90 ↓ |
|---|---|---|---|
| DNO | 99.3 | 0.80 | 1.39 |
| DARTControl | 98.2 | 0.08 | 0.31 |
| AffordMotion (HU) | 43.1 | 0.23 | 0.78 |
| AffordMotion (all) | 37.2 | 0.29 | 1.87 |
| **Ours** $w_s = 0.3$ | 64.0 | 0.28 | 0.99 |
| **Ours** $w_s = 1.0$ | 80.6 | 0.13 | 0.57 |
| **Ours** $w_s = 2.0$ | 89.7 | 0.08 | 0.32 |

Table 6: **Comparison of penetration statistics across methods.** ZR denotes the fraction (%) of penetration-free frames among all evaluation frames (higher is better), while P50 and P90 are the 50th and 90th percentiles of the per-frame maximum penetration depth computed over colliding frames only (lower is better).

representation used in Guo et al. (2022). We therefore pretrain MDM using the following losses:

$$\mathcal{L}_{\text{joints}} = \mathbb{E}_{x_0 \sim q(x_0|\mathcal{T}),\, t \sim [1,T]} \left[ \left\| \text{FK}(x_0) - \text{FK}(\mathcal{D}_\theta(x_t, t, \mathcal{T})) \right\|_2^2 \right], \qquad (6)$$

$$\mathcal{L}_{\text{vel}} = \mathbb{E}_{x_0 \sim q(x_0|\mathcal{T}),\, t \sim [1,T]} \left[ \left\| \text{diff}(\text{FK}(x_0)) - \text{diff}(\text{FK}(\mathcal{D}_\theta(x_t, t, \mathcal{T}))) \right\|_2^2 \right]. \qquad (7)$$

where FK denotes forward kinematics, and diff refers to the temporal difference of the joint positions. The total loss is given by:

$$\mathcal{L} = \mathcal{L}_{\text{t2m}} + \lambda_{\text{joints}} \mathcal{L}_{\text{joints}} + \lambda_{\text{vel}} \mathcal{L}_{\text{vel}}, \qquad (8)$$

where $\lambda_{\text{joints}} = 1$ and $\lambda_{\text{vel}} = 100$.

**Inbetweening Stage.** For the CaKey layers, we use a single-layer MLP with SiLU activations, initialized such that the modulation does not affect the latents at the start of adaptation. Each layer modulates the latents after the self-attention block within each transformer block of MDM. We train for 200k steps with a learning rate of $1 \times 10^{-4}$ using the AdamW optimizer. The same loss functions used in MDM pretraining are applied.

**Scene-Aware Inbetweening Stage.** For the voxel feature extractor, we employ a 512-dimensional ViT with 4 layers and 4 attention heads, using a patch size of 6 to produce 64 patches in total. Scene-conditioning layers are added to all transformer layers of MDM, where cross-attention is applied immediately after the CaKey layers. To stabilize adaptation, we apply layer normalization to both the key–value pairs and the query, and use gradient clipping. Training proceeds for 200k steps with the same loss weights as the text-to-motion stage.

## D  RESULTS

**Distribution of Penetration-Depth.** To provide further detailed performance about scene-awareness, we visualize the comparisons of penetration-depth distribution as shown in Fig 8. Com-

| Method / $w_s$ | R-P (top 3) ↑ | FID ↓ | CFR ↓ | MMP ↓ | JCR ↓ |
|---|---|---|---|---|---|
| point cloud ($w_s = 0$) | 0.803 | 0.395 | 0.297 | 0.447 | 0.306 |
| point cloud ($w_s = 0.3$) | 0.800 | 0.394 | 0.290 | 0.433 | 0.300 |
| point cloud ($w_s = 0.5$) | 0.795 | 0.400 | 0.289 | 0.426 | 0.296 |
| point cloud ($w_s = 2.0$) | 0.667 | 3.93 | 0.275 | 0.397 | 0.280 |
| TSDF ($w_s = 0$) | 0.801 | 0.347 | 0.292 | 0.425 | 0.295 |
| TSDF ($w_s = 0.3$) | 0.786 | 0.545 | 0.259 | 0.347 | 0.257 |
| TSDF ($w_s = 0.5$) | 0.748 | 1.44 | 0.231 | 0.283 | 0.217 |
| TSDF ($w_s = 2.0$) | 0.359 | 20.97 | 0.095 | 0.078 | 0.063 |
| mesh ($w_s = 0$) | 0.801 | 0.471 | 0.311 | 0.503 | 0.333 |
| mesh ($w_s = 0.3$) | 0.800 | 0.489 | 0.305 | 0.486 | 0.327 |
| mesh ($w_s = 0.5$) | 0.798 | 0.508 | 0.302 | 0.475 | 0.322 |
| mesh ($w_s = 2.0$) | 0.747 | 1.47 | 0.267 | 0.381 | 0.274 |
| voxel ($w_s = 0$) | 0.803 | 0.312 | 0.298 | 0.273 | 0.299 |
| voxel ($w_s = 0.3$) | 0.792 | 0.497 | 0.256 | 0.208 | 0.246 |
| voxel ($w_s = 0.5$) | 0.750 | 1.42 | 0.220 | 0.160 | 0.199 |
| voxel ($w_s = 2.0$) | 0.365 | 18.88 | 0.072 | 0.035 | 0.045 |

Table 7: **Effect of different scene representations.**

| Dataset | Motion Semantic | Duration (min) | Scene | Open Source |
|---|---|---|---|---|
| HumanML3D | Diverse | 1715 | ✗ | ✓ |
| HUMANISE | Limited (4 actions) | 600 (purely 51) | ✓ (Synthetic) | ✓ |
| Trumans | Limited (10 actions) | 900 | ✓ | ✓ |
| LaserHuman | Moderate | 180 | ✓ | ✗ |
| SAMP | Limited (5 actions) | 103 | ✗ (Object-only) | ✓ |

Table 8: Comparison of datasets used for scene-aware motion generation.

pared to AffordMotion, our method yields a penetration-depth histogram that is more concentrated near zero, and the bin counts indicate that our motions exhibit fewer penetrations overall. Optimization-based baselines (DNO, DARTControl) also show low penetration, but a closer inspection reveals that this is largely because they generate motions that barely move or exhibit poor adherence to the input text. These tendencies can also be seen quantitatively in Table 6.

**Different Scene Representations.** We further investigate SceneAdapt's robustness to different forms of scene encoding. In addition to our default voxel-based representation, we experiment with three alternative scene encodings: point clouds, TSDF volumes, and meshes. For point clouds, we adopt a Point Transformer encoder; for meshes, a Mesh Transformer encoder; and for TSDF volumes, we reuse the same voxel-ViT encoder employed in our default model. To assess the effect of classifier-free guidance for scene conditioning, we evaluate each representation using multiple scene CFG weights. The full quantitative comparison is reported in Table D. Across all evaluations, we observe that voxel-based scene representations yield the strongest and most stable performance. Their dense and spatially regular structure provides a rich geometric signal that aligns well with SceneAdapt's conditioning architecture, resulting in both high semantic fidelity and strong geometric consistency. TSDF volumes perform competitively, offering similar advantages with slightly smoother geometric fields. In contrast, unstructured or sparse representations such as point clouds and meshes are less effective: although they still enable scene-aware behavior, their irregular sampling and lower spatial density provide weaker geometric cues, limiting their ability to enforce scene constraints.

**Comparison to triplet-based datasets.**

To further support our motivation, we also compare our method with models trained on HUMANISE, a triplet-based dataset, as summarized in Table 10. As shown in the 1st and 2nd rows, models directly fine-tuned on this triplet-based dataset fail to preserve motion semantics. Moreover, models trained from scratch on HUMANISE, listed in the 3rd–5th rows, also underperform. In contrast, our method outperforms all comparison models, thanks to the proposed adaptation strategy.

| Method | Rich Motion Semantics | Scene Geometry Awareness | No Triplet Needed | Learned From GT Interactions | Open Source |
|---|---|---|---|---|---|
| Humanise CVAE Wang et al. (2022) | ✗ | ✓ | ✗ | ✗ | ✓ |
| Arrord Motion Wang et al. (2024a) | ✗ (degrades due to HUMANISE) | ✓ | ✗ | ✗ | ✓ |
| Cen et al. Cen et al. (2024) | ✗ | ✓ | ✗ | ✗ | ✓ |
| TeSMo Yi et al. (2024) | ✗ | ✓ | ✗ | ✓ (interaction) / ✗ (locomotion) | ✓ |
| LaserHuman Cong et al. (2024) | ✗ | ✓ | ✗ | ✓ | ✗ |
| **Ours (SceneAdapt)** | ✓ | ✓ | ✓ | ✓ | ✓ |

Table 9: Comparison of scene-aware motion generation methods.

| Model | R-P (Top3) ↑ | FID ↓ | CFR ↓ | MMP ↓ | JCR ↓ |
|---|---|---|---|---|---|
| MDM + ControlNet | 0.365 | 36.19 | 0.142 | 0.041 | 0.064 |
| MDM + SceneCo Layer | 0.094 | 80.89 | 0.050 | 0.004 | 0.005 |
| HUMANISE (CVAE) | 0.092 | 34.58 | 0.002 | 0.001 | 0.001 |
| AffordMotion (HU) | 0.140 | 21.59 | 0.257 | 0.059 | 0.097 |
| AffordMotion (HU+HML) | 0.305 | 6.320 | 0.429 | 0.254 | 0.321 |
| Ours | 0.792 | 0.497 | 0.256 | 0.208 | 0.246 |

Table 10

# E    POSITIONING

Table 8 highlights a key gap: existing datasets do not jointly provide semantic diversity and scene awareness. This limitation shapes our problem formulation and contributes to the inability of prior methods to generate motions that are both semantically expressive and scene-consistent (Table 9).

# F    USE OF LARGE LANGUAGE MODELS

We only utilized Large Language Models to polish our written draft.

