# OpenReview forum: "SceneAdapt: Scene-aware Adaptation of Human Motion Diffusion"
_ICLR.cc/2026/Conference — Submitted to ICLR 2026_

### Official Review · Reviewer_S7SK · 2025-10-26

**Soundness:** 3
**Presentation:** 3
**Contribution:** 3
**Rating:** 6
**Confidence:** 3

**Summary:**

This paper focuses on HSI and aims to solve the issue of modeling motion semantics and scene-awareness simultaneously. This paper introduces SceneAdapt, a framework that injects scene awareness into text-conditioned motion models by leveraging disjoint scene–motion and text–motion datasets through two adaptation stages. Comprehensive experiments have proven the effectiveness of SceneAdapt.

**Strengths:**

This paper innovatively introduces motion inbetweening into the model to generate scene-aware human motion. The paper is well-developed and provides extensive analysis and visualization results.

**Weaknesses:**

1. The distinction between HSI(Human-Scene Interaction) and scene-aware text-to-motion generation is not clearly stated in this paper.
2. This paper only reports performance on the dataset constructed by the author, and does not conduct experiments on other public datasets.

**Questions:**

1. Why is MDM trained for the task of motion inbetweening instead of directly using the motion inbetweening model?
2. What impact will the absence of Stage 0 have on scene-aware text-to-motion generation performance?
3. Why didn't the author use an autoregressive motion generation model to complete the motion inbetweening task without adding additional modules?

---

> ### Author Response · Authors · 2025-11-25
> **Response to Reviewer S7SK (1/1)**
>
> We thank the reviewer for appreciating our approach as innovative, and our paper well-developed with extensive analysis and visualization results. Here, we address the weaknesses (W) and questions (Q) raised by the reviewer. Please let us know if you have any further questions or require additional results.
>
> ### **[W1] HSI vs scene-aware text-to-motion**
>
> We thank the reviewer for pointing this out. HSI (Human–Scene Interaction) is a broader concept that encompasses diverse tasks, whereas scene-aware text-to-motion generation is a specific task that aims to generate natural human motion consistent with both the scene and the text. We clarify this goal explicitly in General Response 1 and will adjust the paper accordingly.
>
> We also agree that using HSI and scene-aware text-to-motion together can be confusing. Therefore, we have replaced HSI with “scene-aware human motion” throughout the main paper.
>
> ### **[W2] External datasets evaluation**
>
> We thank the reviewer for pointing out the lack of additional test sets. As stated in our General Response 1, our goal is to generate semantically rich motion that is also scene-aware. Evaluating this properly would ideally require a large-scale test set of (text, scene, motion) triplets, which is precisely the type of data that is **extremely costly to collect** and that **our method is designed to avoid**.
>
> Therefore, in order to evaluate our goal, we construct an evaluation set from two large public datasets, HumanML3D and TRUMANS. Our construction protocol pairs diverse HumanML3D text prompts with TRUMANS scenes and starting poses (after SDF-based filtering), yielding a **wide range of semantic descriptions and plausible scene configurations**. Although not perfect as using ground truth (text, scene, motion) triplets, we believe that it provides a reasonable and scalable testbed for comparing methods under a shared setting. We have added additional discussions regardnig our evaluation set in Appendix: Evaluation set construction.
>
>
> ### **[Q1] Why not use an inbetweening-specific model?**
>
> This is because our goal, as stated in General Response 1, is scene-aware text-to-motion generation. Inbetweening models trained specifically for text-to-motion inbetweening (e.g., CondMDI) can generate motion from text plus keyframe inputs, but not from text alone, which is why they are not suitable for direct use as our base text-to-motion model.
>
>
> ### **[Q2] Impact of Stage 0**
>
> Stage 0, the text-to-motion pretraining on a text–motion paired dataset, is crucial for scene-aware text-to-motion generation performance. Stages 1 and 2 focus on injecting scene awareness into this base model while preserving its existing text-to-motion capabilities, and therefore do not aim to further improve the base text-to-motion performance itself. Without Stage 0, our pipeline would be meaningless: the fact that current text-to-motion models such as MDM already excel at generating semantically rich motion is precisely what makes adapting them for scene awareness a reasonable and effective strategy.
>
> ### **[Q3] Using Autoregressive model**
>
> Thank you for your suggestion. While autoregressive models such as DiP excel at generating motion from text, we are not aware of any autoregressive motion models that have been used for the motion inbetweening task. If such models turn out to be effective at inbetweening, it would be reasonable to apply our approach to them as well, rather than only to bidirectional models. Exploring how to infinitely roll out scene-aware motion under continuous text inputs would be an interesting direction for future work.

---

### Official Review · Reviewer_Wz4f · 2025-10-26

**Soundness:** 3
**Presentation:** 2
**Contribution:** 3
**Rating:** 4
**Confidence:** 5

**Summary:**

# Summary
The paper proposes SceneAdapt, a two-stage adaptation pipeline that injects 3D scene awareness into a pretrained text-conditioned motion diffusion model (MDM) using disjoint text–motion and scene–motion datasets.   Stage 1: Insert Context-aware Keyframing (CaKey) layers that modulate only keyframe latents to learn motion inbetweening while preserving the pretrained latent manifold. Stage 2 (scene-aware inbetweening): Freeze CaKey and add scene-conditioning cross-attention layers that query voxel ViT patch embeddings so each frame attends to local scene context;
Once done, during the Inference, one can generate scene-aware motion from text + scene without keyframes, trading off semantics vs geometry via two CFG scales.

**Strengths:**

# Strengths
- Simple, plausible idea: Using inbetweening as a proxy to bridge two datasets is intuitive and effective, avoiding costly text–scene–motion triplets.
- Controllable trade-off: Dual CFG scales for text and scene provide a clear knob to balance semantics and geometry.
- The writing is generally ok but with some problems (See the next section).
- Authors promise code/model release.

**Weaknesses:**

# Weaknesses
- The idea is not very novel. Keyframe-based motion gen and attention-based adaptor have been largely explored.

- Visual comparisons can be clearer; collisions are still visible. In the “selected_blinded” materials, please consider adding method-name overlays on every clip. Also, it's better to make them into one single video. Meanwhile, despite improvements, there are some noticeable penetrations in many cases in the video.

- Overstated positioning in related work. The claim that prior work addresses “either semantics or scene-awareness in isolation” is too strong… Recent efforts *do* combine text and scene: 1) Move as You Say, Interact as You Can: Language-guided Human Motion Generation with Scene Affordance, CVPR 2024. 2) Generating Human Interaction Motions in Scenes with Text Control, ECCV 2024. 3) Generating Human Motion in 3D Scenes from Text Descriptions, CVPR 2024. 4) LaserHuman: Language-guided Scene-aware Human Motion Generation in Free Environment, 2024... The contribution should be framed as *how* SceneAdapt avoids triplets and *how* it injects scene priors efficiently.

- Evaluation-set construction may bias outcomes. The test set pairs HumanML3D text–motion with random TRUMANS start poses/trajectories after SDF-based filtering. This can inadvertently favor or disfavor certain prompts and should be discussed as a limitation (correct me if I am wrong).

- Writing. Several typos/grammar issues (e.g., “exsisting,” “well-adpated,” “indicies,” “sacraficing,” “Qaulitative”)...; terminology for the scene-conditioning module (“SceneCo/ScenCo”) is inconsistent. Also, “et, as these datasets omit scene context, such models remain blind to spatial constraints and cannot generate motions that interact plausibly with the environment” - this is because they are targeting text-to-motion generation…

**Questions:**

1. Could you report **penetration-depth distributions** (e.g., percentiles or max/min) in addition to means? Table 1 (p.7) averages may hide long tails with severe MMP. I did see some penetration cases in the submitted video.

2. Robustness to scene representations: How does performance change with **sparser or noisier reconstructions** (e.g., raw point clouds / TSDF / meshes) vs voxel ViT patches?

**Details Of Ethics Concerns:**

/

---

> ### Author Response · Authors · 2025-11-25
> **Response to Reviewer Wz4f (1/5)**
>
> We thank the reviewer for appreciating our approach as a plausible, effective idea, while providing controllability for text and scene. Besides the general response, we have also incorporated your suggestions into the revised manuscript and provided detailed explanations for each weakness (W) and question (Q) below. Please let us know if you have any further questions or require additional results.
>
> ### **[W1] Novelty**
> We thank the reviewer for the comment on regarding the novelty of the individual components. While it is true that keyframe-based motion generation and attention-based adaptor have been explored individually, our novelty lies  not in the overall concept of each component, but in the specific design and application of their usage to solve a critical and previously unchallenged practical problem in our domain: injecting scene awareness into pre-trained text-to-motion models without relying on expensive text-motion-scene paired datasets. Our main contributions, which we believe offer novelty are the following:
>
> **[Novelty 1: Problem formulation]** While scene-aware text-to-motion generation using text-motion-scene datasets has been explored, **adaptation of text-to-motion models to inject scene-awareness using only motion-scene pairs is not**. This problem formulation allows us to circumbent the infeasiblity of collected data that is not only has rich motion semantics ("playing basketball", "performing martial arts", "dancing like michal jackson"), but also close geometic relation ship with the surrounding scene.
>
> **[Novelty 2: Architectual Design]** While keyframing has been explored, we are the first to explore the **adaptation of text-to-motion models to motion inbetweening**, which requires perserving the latents, while allowing sparse spatial control, which motivates the design choice of CaKey layers. Moreover, we believe that non other works tries to use adaptors that **spatio-temporally aligns motion and scene using cross attention**.
>
> **[Novelty 3: Performance]** While previous approaches fail to generate motion that is both semantically rich and collision-free, we show, both qualitatively and quantitatively, that SceneAdapt outperforms prior methods **not only in geometric scene awareness but also in text adherence to diverse prompts**.
>
>
> We hope this clarifies how our contribution goes beyond reusing existing components, by **formulating a new adaptation setting** and **designing architectures** specifically tailored to this problem, with **measurable benefits** in practice.

---

> ### Author Response · Authors · 2025-11-25
> **Response to Reviewer Wz4f (2/5)**
>
> ### **[W3] Positioning**
>
> Thank you for pointing this out. We agree that our original phrasing “existing motion generation approaches address either semantics or scene-awareness in isolation” was too strong and could mislead readers, especially given the recent emergence of text-and-scene–conditioned works such as [1,2,3,4,5].
> To avoid overstatement and false claims, we first organize our position compared to related works as shown below.
> | Method                | Rich Motion Semantics | Scene Geometry Awareness | No Triplet Needed | Learned From Ground Truth Interactions |
> |-----------------------|-----------------------|---------------------------|-------------------|----------------------------------------|
> | Humanise CVAE [1]     | ✘                     | ✔️                        | ✘                 | ✘                                      |
> | Arrord Motion [2]     | ✘ (degrades due to HUMANISE) | ✔️                 | ✘                 | ✘                                      |
> | Cen et al. [3]        | ✘                     | ✔️                        | ✘                 | ✘                                      |
> | TeSMo [4]             | ✘                     | ✔️                        | ✘                 | ✔️ (interaction) / ✘ (locomotion)      |
> | LaserHuman [5]        | ✘                     | ✔️                        | ✘                 | ✔️                                      |
> | **Ours (SceneAdapt)** | **✔️**                | **✔️**                    | **✔️**            | **✔️**                                  |
>
>
> Our approach is the only method that can effectively produce human motion that are semantically rich (adhering to diverse text-prompts) that does not collide with the given scene. Therefore, we rephrase our statement as
>
> “However, existing motion generation approaches fail to generate diverse motion while simultaneously respecting scene constraints, since constructing large-scale datasets with both rich text-motion coverage and precise scene interactions is extremely challenging.”
>
> Moreover, as the reviewer pointed out, scene-aware text-to-motion generation has been explored before, but not in the context of avoiding triplet data. We have revised the paper to emphasize that our focus is on adapting pretrained text-to-motion models without relying on text–scene–motion triplets.
>
> [1] HUMANISE: Language-conditioned Human Motion Generation in 3D Scenes, Wang et al., 2022
>
> [2] Move as You Say, Interact as You Can: Language-guided Human Motion Generation with Scene Affordance, Wang et al., 2024
>
> [3] Generating Human Motion in 3D Scenes from Text Descriptions, Cen et al., 2024
>
> [4] Generating Human Interaction Motions in Scenes with Text Control, Yi et al., 2024
>
> [5] LaserHuman: Language-guided Scene-aware Human Motion Generation in Free Environment, Cong et al., 2024

---

> ### Author Response · Authors · 2025-11-25
> **Response to Reviewer Wz4f (3/5)**
>
> ### **[Q1] Penetration-depth Distribution**
>
> The reviewer raised an important point about severe maximum penetration values being hidden by averaging. To address this more intuitively, we now include histograms of maximum penetration values in Figure 6 (Appendix: Results - we have added it to the appendix.) of the revised manuscript. The results show that SceneAdapt exhibits fewer penetrations overall, and while optimization-based methods (DNO, DARTControl) achieve exceptionally low penetration (due to iterative optimization with respect to scene collisions), they suffer from low text adherence and often generate motions that are largely static (see Table 1, Figure 4 main paper).
>
> We also report the 50th and 90th percentiles of the per-frame maximum penetration depth for each model, as shown in below table. As explained in Appendix: Results, frames without any penetration are excluded when computing these percentiles. Therefore, we additionally report the fraction of penetration-free frames among all evaluation frames, denoted as the zero-ratio (ZR). **For a fair comparison**, we hope the reviewers will consider both the percentile statistics and the ZR values.
>
> ** P50, P90: lower is better; ZR: higher is better.
>
> [Table D]
>
> | Method             | ZR (%) | P50   | P90  |
> |:------------------ | --- |:----- |:---- |
> | DNO                |  99.3   | 0.80 | 1.39 |
> | DARTControl        |  98.2   | 0.08 | 0.31 |
> | AffordMotion (HU)  |  43.1   | 0.23  | 0.78 |
> | AffordMotion (all) |  37.2   | 0.29  | 1.87 |
> | **Ours**  $w_s$ = 0.3       |  64.0   | 0.28  | 0.99 |
> | **Ours**  $w_s$ = 1.0            |  80.6   | 0.13  | 0.57 |
> | **Ours**  $w_s$ = 2.0        |  89.7   | 0.08  | 0.32 |

---

> ### Author Response · Authors · 2025-11-25
> **Response to Reviewer Wz4f (4/5)**
>
> ### **[Q2] Robustness to different scene encodings**
>
>
> We thank the reviewer for the insightful question regarding SceneAdapt’s robustness to different scene encodings. To evaluate this, we trained SceneAdapt using four alternative scene representations: point clouds, TSDF volumes, meshes, and voxel patches (our default). For point clouds, we used a Point Transformer; for meshes, a Mesh Transformer; and for TSDF, the same voxel-ViT encoder as our default voxel model.
>
> The full quantitative comparison across different scene classifier-free guidance weights ($w_s$ ) is shown below:
> | Method / $w_s$          | R-P (top 3) ↑ | FID ↓   | CFR ↓  | MMP ↓  | JCR ↓  |
> |------------------------|---------------|---------|--------|--------|--------|
> | point cloud ($w_s$= 0)   | 0.803         | 0.395   | 0.297  | 0.447  | 0.306  |
> | point cloud ($w_s$ = 0.3) | 0.800         | 0.394   | 0.290  | 0.433  | 0.300  |
> | point cloud ($w_s$ = 0.5) | 0.795         | 0.400   | 0.289  | 0.426  | 0.296  |
> | point cloud ($w_s$ = 2.0) | 0.667         | 3.93    | 0.275  | 0.397  | 0.280  |
> | TSDF ($w_s$ = 0)          | 0.801         | 0.347   | 0.292  | 0.425  | 0.295  |
> | TSDF ($w_s$ = 0.3)        | 0.786         | 0.545   | 0.259  | 0.347  | 0.257  |
> | TSDF ($w_s$ = 0.5)        | 0.748         | 1.44    | 0.231  | 0.283  | 0.217  |
> | TSDF ($w_s$ = 2.0)        | 0.359         | 20.97   | 0.095  | 0.078  | 0.063  |
> | mesh ($w_s$ = 0)          | 0.801         | 0.471   | 0.311  | 0.503  | 0.333  |
> | mesh ($w_s$ = 0.3)        | 0.800         | 0.489   | 0.305  | 0.486  | 0.327  |
> | mesh ($w_s$ = 0.5)        | 0.798         | 0.508   | 0.302  | 0.475  | 0.322  |
> | mesh ($w_s$= 2.0)        | 0.747         | 1.47    | 0.267  | 0.381  | 0.274  |
> | voxel ($w_s$ = 0)         | 0.803         | 0.312   | 0.298  | 0.273  | 0.299  |
> | voxel ($w_s$ = 0.3)       | 0.792         | 0.497   | 0.256  | 0.208  | 0.246  |
> | voxel ($w_s$ = 0.5)       | 0.750         | 1.42    | 0.220  | 0.160  | 0.199  |
> | voxel ($w_s$ = 2.0)       | 0.365         | 18.88   | 0.072  | 0.035  | 0.045  |
>
> Overall, we observe three consistent trends:
>
> **1) Voxel representations yield the strongest and most stable performance.**
> Their regular grid structure provides a clean, explicit encoding of the scene that integrates well into SceneAdapt’s conditioning pipeline. This is consistent with recent works that use voxels as scene representation.
>
> **2) TSDF perform competitively with voxels.**
>
> **3) Sparser or unstructured encodings (point clouds, meshes) are less effective.**
> While point clouds and meshes still enable reasonable scene-aware behavior, their lack of regular structure and reduced spatial density make them less informative for enforcing geometric constraints.
>
> In summary, SceneAdapt works across a broad range of scene encodings, but voxel-based and TSDF encodings provide the strongest geometric signal, likely due to their dense, spatially regular structure. We will include these results in the revision to highlight SceneAdapt’s robustness and clarify the impact of scene representation choice.

---

> ### Author Response · Authors · 2025-11-25
> **Response to Reviewer Wz4f (5/5)**
>
> ### **[W4] Evaluation set can bias outcomes**
>
> We acknowledge that our evaluation pipeline may inherently favor or disadvantage certain text prompts. Although we avoid extreme cases, such as pairing a prompt requiring forward motion with a scene where the character begins directly in front of a wall, by applying SDF-based filtering, this filtering primarily ensures feasibility rather than perfect neutrality. For example, a prompt like “a person running” will naturally be easier to satisfy without penetrations when matched to a large room compared to a smaller one. Nonetheless, this approach allows us to **systematically evaluate a broad range of text prompts and scene-awareness behaviors**, as discussed in our Global Response (Primary Goal of SceneAdapt), without relying on a real-world scene–text–motion triplet test set with rich and diverse semantics, which would be **extremely difficult to collect in practice**. We have added this discussion on the limitation.
>
> ### **[W2] Visual Comparisons**
>
> We thank the reviewer for the feedback regarding the visual results. We have added clearer visual comparisons to our anonymous project page (https://zaq1xsw2-cde3.github.io/iclr2026_rebuttal/). While SceneAdapt is not perfect and may still exhibit occasional collisions, particularly in cluttered scenes, we would like to highlight that our quantitative results (Table 1 in the main paper) show that increasing the scene CFG weight substantially reduces collisions.
>
> To further illustrate this effect, we additionally provide visual examples generated with varying scene CFG weights. As the scene CFG weight increases, scene penetration consistently decreases, demonstrating that our model can be **steered toward safer, more physically plausible motions**.
>
>
> ### **[W5] Writing**
>
> All reported typos, naming inconsistencies and clarity issues have been fixed in the revised manuscript. We have changed "Yet, as these datasets omit scene context, such models ~" to "Yet, as these models only target text-to-motion, such models ~". Thank you for your feedback.

---

### Official Review · Reviewer_ey96 · 2025-10-30

**Soundness:** 3
**Presentation:** 3
**Contribution:** 2
**Rating:** 4
**Confidence:** 3

**Summary:**

This paper introduces SceneAdapt, a two-stage adaptation framework that effectively injects scene awareness into a pretrained text-conditioned motion diffusion model by leveraging motion inbetweening as a proxy task, enabling the generation of semantically rich and physically plausible human motions without requiring costly text-scene-motion triplets.

**Strengths:**

**Practical Problem Formulation**
- Solves the critical problem of injecting scene awareness into pretrained text-to-motion models
- Avoids costly collection of text-scene-motion triplets by leveraging existing disjoint datasets

 **Well-Designed Two-Stage Framework**
- **Stage 1**: Learns motion inbetweening while preserving the pretrained model's capabilities
- **Stage 2**: Focuses exclusively on scene awareness building on a stable motion foundation
- Clear separation of concerns prevents catastrophic forgetting

 **Novel CaKey Layer Design**
- **Sparse modulation** only affects keyframe latents, preserving the generative manifold
- **Context-aware** through diffusion timestep and self-attention inputs
- Superior to standard adaptation methods like LoRA for this specific task

 **Advanced Scene Conditioning**
- Uses **patch-wise cross-attention** instead of global scene vectors
- Enables dynamic, localized interaction with scene geometry over time
- More physically plausible than previous global conditioning approaches

**Weaknesses:**

**Missing Critical Comparison with Triplet Datasets**
- **Core Issue**: Fails to compare against models trained on existing **text-scene-motion triplet datasets**
- **Consequences**: Cannot determine if performance stems from:
  - The **adaptation method itself**, OR
  - Simply **inheriting advantages** from larger text-motion pre-training data
- **Relevant References**:
  - `[HUMANISE]` Wang et al., "Humanise: Language-conditioned Human Motion Generation in 3D Scenes", NeurIPS 2022
  - `[LaserHuman]` Li et al., "LaserHuman: Language-guided Scene-aware Human Motion Generation", arXiv 2024

**Overly Narrow Definition of "Scene Awareness"**
- **Core Issue**: Equates scene awareness primarily with **collision avoidance metrics** (CFR, MMP, JCR)
- **Consequences**: Shows no capability for **positive, goal-oriented interactions**

**Unverified Proxy Task Justification**
- **Core Issue**: Entire pipeline relies on unsubstantiated assumption that **motion inbetweening** is optimal proxy for scene awareness
- **Consequences**: No theoretical/empirical evidence why temporal interpolation enables spatial reasoning

The adaptation paradigm cannot be properly evaluated without fair comparisons against models trained on existing triplet datasets.

**Questions:**

**Question 1:** The evaluation set is constructed by randomly pairing HumanML3D texts with TRUMANS scene trajectories, creating semantically mismatched pairs (e.g., "a person does a cartwheel" in a narrow hallway). How can we trust that the reported R-Precision and FID scores accurately reflect model performance when the ground truth pairs themselves are semantically implausible? Would results on a curated, semantically consistent subset tell a different story?

**Question 2:** Given the existence of text-scene-motion triplet datasets like HUMANISE and LaserHuman, why wasn't SceneAdapt compared against models trained end-to-end on such data? Without this comparison, how can we determine whether your performance advantage comes from the adaptation method itself, or simply from leveraging a larger and richer text-motion pre-training dataset?

**Question 3:** Your method demonstrates competence in collision avoidance but shows no capability for positive object interactions (e.g., sitting on chairs, grabbing cups). Do you consider your current approach fundamentally limited to passive obstacle avoidance, or can it be extended to support goal-directed interactions? What architectural changes would be needed to handle such "functional" scene awareness?

---

> ### Author Response · Authors · 2025-11-25
> **Response to Reviewer ey96 (1/2)**
>
> We thank the reviewer for appreciating our problem formation as practical, and our method as well-designed, novel and advanced. Here, we address the weaknesses (W) and questions (Q) raised by the reviewer. Please let us know if you have any further questions or require additional results.
>
> ### **[Q1] Validity of the evaluation set**
>
> Thank you for raising concerns on the validity of our evaluation set. The reviewer is correct in that there could be mismatched pairs such as "a person doing cartwheel" in a narrow hallway. This is the exact reason why we filtered motion starting points in the TRUMANS trajectories that are too close to the surrounding scenes. As noted in Appendix: Evaluation Set Construction, we filtered these positions using SDF values as they can implausible synthesis, and degrade our evaluation experiments. As a result, our evaluation set accuratly evaluates our goals stated in our Global Response 1 (Primary Goal of SceneAdapt).
>
>  Curating a real-world scene-text-motion triplet test set would be extremely costly, though it would likely overcome the trade-offs currently observed between motion generation metrics (FID, R-P) and scene-awareness metrics (CFR, MMP, JCR). Despite this limitation, our current evaluation set is sufficiently effective to validate the primary goals of this paper.
>
>
> ### **[W1, Q2] Comparison to triplet-based datasets**
>
> We thank the reviewer for this important suggestion. To directly address this concern, we added baselines that adapt MDM on text-scene-motion data as well as models trained end-to-end on them.
>
> 1) MDM + ControlNet adaptation using HUMANISE
> 2) MDM + SceneCo adaptation using HUMANISE
> 3) HUMANISE cvae
>
> the results are shown below
>
>
> [Table C]
> | Model                        | R-P (Top3) | FID  |  CFR   |  MMP   |  JCR   |
> | ---------------------------- | ---------- | ---- | --- | --- | --- |
> | (1) MDM + ControlNet             | 0.365      | 36.19  | 0.142    | 0.041    | 0.064    |
> | (2) MDM + ScenCo Layer (Sec 3.3) | 0.094       | 80.89 | 0.050    |  0.004   | 0.005    |
> | (3) HUMANISE cvae|  0.092   | 34.58 | 0.002   | 0.001   | 0.001  |
> | afford motion (HU)| 0.140    | 21.59 | 0.257  | 0.059   | 0.097  |
> | afford motion (HU + HML) | 0.305|   6.320 | 0.429 | 0.254   | 0.321   |
> | ours $w_s$ =0.3 |  0.792   | 0.497 | 0.256   | 0.208   | 0.246  |
>
> To implement (1), we use a ViT-based voxel encoder together with a ControlNet-style conditioning module, following the design principles of OmniControl. For (2) we use only the SceneCo layer on top of MDM. Both models were trained using HUMANISE triplet dataset, NOT using our adaptation scheme.
>
> **Using HUMISE to adapt MDM degrades motion semantics**
>
> As shown in the table, both adaptation baselines (1),(2) underperform SceneAdapt in text alignment. These results support the core motivation of our method: even when starting from a text-rich model pretrained on HumanML3D, directly adapting on current triplet datasets leads to semantic collapse due to their limited scale and restricted text/motion diversity. This limitation is also evident in Fig. 1a, where HUMANISE and TRUMANS occupy a much narrower motion embedding distribution than HumanML3D. To be specific, HUMANISE provides text annotations for **only four action categories**: walk, sit down, stand up, and lie down. The latter three are nearly stationary, exhibiting no root-position translation, and together they account for roughly twice as many sequences as the walking category. Thus, **naive finetuning using current triplet datasets like HUMAISE is not effective for our goal**.
>
> **Using HUMANISE for end-to-end training lack performance**
>
> We report metrics with HUMANISE cvae and AffordMotion (trained on HUMANISE alone and on HUMANISE+HML3D) as a baseline in Table C. Note that AffordMotion is on main table 1 as well. Although AffordMotion represents the strongest available model that is trained on text–scene–motion triplet dataset, SceneAdapt surpasses it in both text fidelity and scene consistency. This shows how **using accurate scene-motion data without text can be more powerfull** than using synthetic triplet data with a narrow range of motion semantics.
>
>
> Collectively, these new experiments and existing comparisons demonstrate that SceneAdapt provides capabilities that models trained on current synthetic triplet datasets cannot achieve, even when adapted from the same pretrained checkpoint.

---

> ### Author Response · Authors · 2025-11-25
> **Response to Reviewer ey96 (2/2)**
>
> ### **[W2, Q3] Narrow definition of “scene awareness”**
>
> As stated in our Global Response (Primary Goal of SceneAdapt), our objective is to generate geometric scene-aware motion with rich motion semantics. However, we do agree that functional behaviors (e.g., sitting, reaching, grasping) are also an important dimension of scene awareness that goes beyond collision avoidance.
>
> Therefore, to examine whether our approach can support such behaviors, we explored conditioning the final model on sparse end-of-motion keyframes (interpretable as goal poses), text and scene. As our last stage adaptation strategy is scene-aware motion inbetweening, we find that the model can indeed produce goal-directed motions under this setting.
>
> As shown in the **anonymous website**(https://zaq1xsw2-cde3.github.io/iclr2026_rebuttal/), the model is able to sit on chairs and reach toward objects across diverse scenes when given a goal pose, scene, and text description. Utilizing scene-aware pose generation methods to guide SceneAdapt toward functional, goal-directed motion seems like a promising direction for future work.
>
>
>
> ### **[W3] Justification of inbetweening as proxy**
> While **we do not claim that motion inbetweening is the unique or theoretically optimal proxy task**, we view it as a practical and empirically effective choice in our setting.
>
> (1) As shown in Fig. 5(d), removing the inbetweening objective leads to degraded motion semantics, whereas our adaptation paradigm preserves rich motion semantics. In addition, directly training the model on the triplet dataset leads to worse performance, as shown in the accompanying table (see the results of “MDM + ControlNet” and “MDM + SceneCo layer”). In short, these result indicate that leveraging the proxy task to avoid semantic collapse is necessary. Therefore, we opt to design this proxy task as motion inbetweening and validate its effectiveness empirically.
>
> (2) From a dataset perspective, the only modality shared between large text-to-motion datasets (e.g., HML3D ; no scene) and human-scene interaction datasets (e.g., HUMANISE, TRUMANS ; narrow semantic distribution of text)is motion. Motion inbetweening is attractive choice because it uses only motion as supervision, making it directly applicable to both datasets.
>
> While exploring alternative proxy task is a promising direction for future work, we believe that motion inbetweening is a sound and validated proxy task.

---

### Official Review · Reviewer_qMbQ · 2025-10-31

**Soundness:** 3
**Presentation:** 3
**Contribution:** 2
**Rating:** 4
**Confidence:** 4

**Summary:**

This paper proposes a two-stage adaptation framework named SceneAdapt, designed to inject scene awareness into pre-trained text-to-motion diffusion models. The core idea is to cleverly leverage "motion inbetweening" as a text-free proxy task to bridge two disjoint datasets: a large-scale text-motion dataset and a smaller scene-motion dataset. In the first stage, the model learns inbetweening via a novel CaKey layer; in the second stage, it learns to perform inbetweening within a scene via a SceneCo layer, thereby acquiring scene awareness. Ultimately, the model is capable of generating motions that are both consistent with the text description and physically plausible within the given scene.

**Strengths:**

1.  Leveraging "motion inbetweening" as a proxy task to decouple and bridge two heterogeneous datasets is a very clever idea. It provides an effective new paradigm for multi-modal model fusion in data-scarce scenarios.
2.  The method directly confronts the bottleneck of data acquisition by proposing a solution that does not rely on expensive triplet-annotated data. This makes it more feasible to imbue existing large-scale motion generation models with scene awareness.
3.  The designs of the CaKey and SceneCo layers are well-targeted. Specifically, the CaKey layer's use of sparse modulation to preserve the latent space of the pre-trained model and the SceneCo layer's use of cross-attention to focus on local scene features both demonstrate sound design principles.

**Weaknesses:**

The evaluation protocol is biased: Due to the lack of a real-world test set, the authors construct an evaluation set by randomly matching text-motion pairs with scenes. This is a significant flaw. This approach can generate a large number of illogical test cases (e.g., testing the instruction "walks up the stairs" in a room with no stairs). Consequently, the current evaluation metrics primarily measure the model's geometric collision avoidance capabilities when performing an arbitrary action in an arbitrary scene, rather than its ability to generate a reasonable action in a logically matched scene. This diminishes the persuasiveness of the evaluation results.

**Questions:**

1.  Regarding the validity of the evaluation set: How do the authors address the potential for logical mismatches arising from the "random matching" of the evaluation set? For instance, when the text description is entirely unrelated to the scene content, how should we interpret the model's performance and the meaning of metrics like R-Precision?
2.  Regarding the potential bias of inbetweening: Is it possible that using motion inbetweening as the core proxy task might inadvertently bias the model towards generating smooth, continuous motions, thereby affecting its ability to generate more explosive or abrupt actions?

---

> ### Author Response · Authors · 2025-11-25
> **Response to Reviewer qMbQ (1/1)**
>
> We thank the reviewer for appreciating our approach as a clever solution that does not require triplet data, and a soundly, well designed method. Here, we address the weaknesses (W) and questions (Q) raised by the reviewer. Please let us know if you have any further questions or require additional results.
>
> ### **[W1,Q1] : Validity of the evaluation set**
>
> Thank you for raising concerns on the validity of our evaluation set. As our test set is constructed using HumanML3D texts, which are not scene-aware, scene-related texts rarely exist, except for some special cases such as the example the reviewer mentioned ("walks up the stairs"). As noted in the Appendix: Evaluation Set Construction, **we excluded such text-motion pairs, ensuring that such scene-related text does not exist in our test set.**
>
>
> Moreover, as stated in our Global Response 1 (Primary Goal of SceneAdapt), our objective is to generate **semantically rich motion** that maintains **geometric scene-awareness** (i.e., avoids scene penetration).  Therefore, our evaluation is designed to answer the following question:
>
> > **"How natural (FID), text-adhering (R-P), and non-colliding (CFR, MMP, JCR) are the generated motions, given semantic text (from HumanML3D) and geometric constraints (from TRUMANS)?"**
>
> In this setting, our evaluation protocol **does not require** and **does not utilize** text prompts that contain explicit scene vocabulary (e.g., "refrigerator," "stairs," or "table"). The model is tested purely on its ability to satisfy a **semantic text condition** and a **geometric collision constraint** simultaneously, justifying our evaluation set construction.
>
>
> ### **[Q2] Bias toward smooth motions due to inbetweening**
>
> We thank the reviewer for pointing out potential flaws in our inbetweening adaptation strategy. To investigate this concern, we calculated the average magnitude of **velocity** and **acceleration** for the generated motions, comparing our adapted model against our base text-to-motion model.
>
> [Table B]
>
> | Model | Average Velocity | Average Acceleration | FID |
> | :--- | :--- | :--- |:---|
> | **MDM** | 0.0133 | 0.00313 |0.479|
> | **Adapted (ws=0.0)** | 0.0112 | 0.00300 |0.312|
>
> Although the adapted model showed a small decrease in average velocity and acceleration, the percentage drop was **trivial**. More importantly, as demonstrated in **Table 1** of our main paper, the adapted model shows an **improved FID** score, which indicates that the generated motions are **more natural** than those produced by the base model.

---

> > ### Comment · Reviewer_qMbQ · 2025-11-28
> > **Reply to rebuttal**
> >
> > Thank you to the authors for the detailed reply. The answer resolved my concerns, and the article's method of connecting two different datasets is a good complement to the current situation, where datasets simultaneously encompassing text comprehension and scene perception are lacking. I am willing to raise the score to 6 points.

---

### Author Response · Authors · 2025-11-25
**General Response 1**

We thank all the reviewers for their comments and suggestions to strengthen our work. They described our use of motion inbetweening to bridge disjoint text–motion and scene–motion datasets as a clever and intuitive idea (qMbQ, ey96, Wz4f, S7SK), our formulation as a practical way to inject scene awareness without costly text–scene–motion triplets (qMbQ, ey96, Wz4f), and our two-stage CaKey/SceneCo design as well-targeted and well-designed (qMbQ, ey96). They also characterized the overall approach as innovative and effective, supported by extensive analysis and visualizations (Wz4f, S7SK).

To provide reviewers with **additional qualitative results**, we have created an anonymous website containing extensive comparisons between prior works and SceneAdapt.

**Anonymous website link**: https://zaq1xsw2-cde3.github.io/iclr2026_rebuttal/

To foster constructive discussion, we next **clarify the primary goal of SceneAdapt** and then respond to individual comments.


---

### Primary Goal of SceneAdapt

Our goal is to train a text-to-motion model that is both:

1.  **Semantically rich:** Generating motions with diverse semantics ranging from low-level (e.g., "a person walks forward") to high-level behaviors (e.g., "a person dancing like a chicken", "a person doing martial arts").
2.  **Scene-aware:** Specifically, we refer to *geometric* scene awareness, where the model must generate motion that does not collide with the given scene geometry. Note that while some works address scene-awareness as a combination of "*scene semantic awareness*" (e.g., walking to a specific chair) and "*scene geometric awareness*" (e.g., not walking through a wall), we specifically tackle the latter.

To achieve both qualities, the most naive solution would be to train a model using a dataset containing (text, scene, motion) triplets with diverse motion semantics and accurate scene-motion relationships. However, as shown in the table below, no such dataset exists (creating one would be infeasible to scale), which is why prior works fail to generate motion that are both semantically rich and scene geometry aware.

[Table A]
| Dataset      | Motion Semantic        | Duration (min)     | Scene              | Open Source |
|------------|-------------------------|---------------------|---------------------|-------------|
| HumanML3D    | Diverse                 | 1715                | ✗                   | ✓           |
| HUMANISE     | Limited (4 actions)     | 600 (purely 51)     | ✓ (Synthetic)       | ✓           |
| Trumans      | Limited (10 actions)    | 900                 | ✓                   | ✓           |
| LaserHuman   | Moderate                | 180                 | ✓                   | ✗           |
| SAMP         | Limited (5 actions)     | 103                 | ✗ (Object-only)     | ✓           |


Therefore, SceneAdapt circumvents this problem by leveraging motion-scene pairs to inject geometric scene awareness into a pre-trained text-to-motion model, enabling scene-aware text-to-motion generation with diverse motion semantics.

---

### Comment · Area_Chair_Snej · 2025-11-26

Dear reviewers,

The authors have responded. We kindly ask you to review the authors' responses to your comments and provide your feedback. Thank you.

Best,

AC

---

### Author Response · Authors · 2025-12-03
**Summary of Discussion Phase**

Dear Area Chairs, Senior Area Chairs and Program Chairs,

We sincerely appreciate your time and effort in handling our submission. We also thank all reviewers for their thoughtful assessments and for recognizing the practicality and novel contributions of SceneAdapt in making text-to-motion models scene-aware.

Below is a summary of the main discussion points during the rebuttal period:

### **1. Evaluation**
Reviewers raised concerns regarding our evaluation set. In the rebuttal, we (1) clarified the **primary goal** of SceneAdapt and (2) reiterated that **special cases are filtered out**. As reflected in reviewer qMbQ’s follow-up comments, these alleviated the reviewer's concern. Moreover, in the revised paper, we (1) conducted and added **user studies**, (2) included **discussions on potential bias** arising from specific prompts, and (3) provided the **distribution of penetration events to illustrate long-tail behaviors**.

### **2. Additional Experiments**
Reviewers requested additional experiments to address several concerns about SceneAdapt. In the rebuttal, we (1) demonstrated that the inbetweening process does **not introduce smoothing**, (2) compared SceneAdapt against **new baselines**, (3) showed its compatibility with **alternative scene representations**, and (4) showed that by interpreting goal positions as very sparse keyframes, we can **generate functional motions** when such goal positions are provided. These additional experiments directly address the issues reviewers raised or were curious about, and we believe the concerns have now been resolved. All corresponding results have been incorporated into the revised paper.

### **3. Novelty and Positioning**
Reviewer Wz4f raised questions regarding SceneAdapt’s novelty and positioning. In our rebuttal, we (1) clarified the **primary goal of our work**, (2) articulated the **novelty** of our approach, and (3) provided **direct comparisons with prior methods**. We have revised the paper to ensure that future readers clearly understand our positioning.

We have incorporated all clarifications, new analyses, and updated results into the revision and highlighted in blue. With these improvements, we believe the paper is significantly strengthened and presents a clearer and more rigorous contribution.

Thank you again for taking the time to read our summary.

Best regards,
The Authors

---

> ### Author Response · Authors · 2025-12-03
> **Revision summary**
>
> ## Revision details.
>
> In the latest version of the paper, we highlighted the revised sentences in blue.
>
> ### Major revisions include
> * A new paragraph on goal-position-conditioned generation
>     - Section 3.4: In response to reviewer ey96, we add a paragraph showing that our method can be extended to goal-position-conditioned generation.
> - Figure 5: To support the above paragraph, we provide additional qualitative results in this visualization.
>
> * Revisions to related work – Human–Scene Interaction (HSI) synthesis
>     - Section 2.1: As reviewer S7SK pointed out, we clarify the goal of the scene-aware text-to-motion generation task in the paragraph on HSI. In addition, we replace the term “HSI” with “scene-aware text-to-motion generation” to avoid confusion. These revisions should help reviewers and future readers better understand the task setting.
>
> * New paragraphs describing additional experiments
>     - Appendix C: This section now includes training details for (1) the MDM pretraining stage, (2) the motion in-betweening stage, and (3) the scene-aware motion in-betweening stage.
>     - Appendix D: To provide a more detailed performance analysis, we add paragraphs on (1) the penetration-depth distribution and (2) robustness to different scene modalities (3) comparisons to models trained on triplet-based dataset. We also visualize the penetration-depth distribution. These revisions resolve the concern raised by reviewer Wz4f, ey96 and may also be helpful for future readers.
>     - Appendix E: Although we already responded to reviewer Wz4f’s concern about the positioning of our work, we further emphasize our contribution in this section by comparing (1) datasets and (2) key characteristics of prior methods.
> * We describe the limitations of our method in the Conclusion section.
> * We revise a sentence in the first paragraph of the Introduction section.
>     - As reviewer Wz4f pointed out, we acknowledge that our previous statement was too strong, so we revise it to use a softer tone while preserving our original intent.
> * We add a table summarizing the user study in the Experiments section.
>     - We report the preference ratios measured during the user study, providing an additional criterion for comparison.
>
> ### Minor revisions include
> - Correcting typos
>     - Typos pointed out by reviewer Wz4f: “exsisting” → “existing”, “well-adpated” → “well-adapted”, “indicies” → “indices”, “sacraficing” → “sacrificing”, “Qaulitative” → “Qualitative”.
>     - Other typos: “perserve” → “preserve”, “capabilites” → “capabilities”, etc.
>
> - Adding an anonymous website for better visualization
>     - As suggested by reviewer Wz4f, we add an anonymous project website to provide clearer visualizations.
>     - Providing an explanation of example cases of the scene-aware text-to-motion task in the Introduction section.

---

> > ### Author Response · Authors · 2025-12-03
> > **Additional User Study Results**
> >
> > ## User Study
> >
> > We conducted a user study with **28 participants**.
> > For each evaluation sequence, we generated motions using four methods (MDM, Afford-Motion, DNO, and ours) and presented them in random order along with the corresponding text prompt.
> > Participants were asked two questions:
> > 1. **Which motion best matches the text description?** (text adherence)
> > 2. **Which motion best avoids collisions with the scene and obstacles?** (collision avoidance)
> >
> > For each question, participants selected a single best motion.
> > We report, for each method, the **preference rate**, defined as the fraction of trials in which the method was chosen (higher is better).
> >
> > **Results**
> >
> > | Method              | Text adherence ↑ | Collision avoidance ↑ |
> > | ------------------- | ---------------- | ---------------------- |
> > | MDM                 | 0.147            | 0.040                 |
> > | Afford-Motion (All) | 0.024            | 0.016                 |
> > | DNO                 | 0.040            | 0.119                 |
> > | **Ours**            | **0.790**        | **0.825**             |
> >
> > These results show that participants overwhelmingly preferred SceneAdapt, both for matching the intended text and for avoiding scene collisions.
> > We believe this user study meaningfully complements our quantitative evaluation and provides further evidence of SceneAdapt’s scene-awareness beyond our curated test set.

---

### Meta-Review · Area_Chair_UtNP · 2026-01-07

**Summary:**

This paper proposes a two-stage adaptation framework for scene-aware text-to-motion generation without relying on text–scene–motion triplets, using motion inbetweening as a proxy task. All reviewers raised concerns about the evaluation protocol, which relies on a constructed test set formed by randomly pairing text–motion samples with scenes and may introduce semantic mismatch and bias. The reported metrics primarily measure collision avoidance and do not fully demonstrate meaningful or functional scene awareness.
Reviewers also questioned the novelty, noting that while the overall pipeline is well engineered, many components build on existing ideas. The authors provided a rebuttal with additional experiments, baselines, and clarifications, which improved the presentation and addressed several implementation-level questions. Nevertheless, the responses largely justify the chosen design rather than fully resolving the core concerns about evaluation validity and novelty. I recommend the authors further strengthen the evaluation and clarify the novelty before resubmitting to a future venue.

**Reviewer Concerns:**

The rebuttal addresses reviewer qMbQ’s main concerns. The reviewer explicitly states that these answers resolve the issues and confirms a willingness to raise the score.

Reviewer ey96 raised concerns about the mismatched evaluation set, the lack of comparisons to triplet-trained models, the narrow definition of scene awareness, and the unproven choice of inbetweening as a proxy task. The rebuttal provided new analysis and experiments. Reviewer ey96’s concerns were partially addressed.

Reviewer Wz4f raised concerns about limited novelty, unclear positioning versus recent scene-aware models, inadequate qualitative evaluation, insufficient penetration statistics, robustness to scene representations, questionable evaluation bias, and writing issues. The rebuttal added new experiments and clarifications. The major concern about novelty is not fully addressed.

Reviewer S7SK raised concerns about unclear terminology, limited evaluation beyond the constructed dataset, and the choice of adaptation strategy over inbetweening or autoregressive baselines. The rebuttal clarifies the distinction between HSI and scene-aware text-to-motion and confirms the paper now adopts consistent terminology. The authors explained why collecting text–scene–motion triplets is impractical and the constructed HumanML3D+TRUMANS setup is a reasonable alternative. But the concerns seem not addressed.

**Reviewer Scores:**

Reviewer qMbQ stated that the rebuttal resolved all concerns and is willing to raise the score.

The rebuttal added new baselines, clarified evaluation validity, demonstrated proxy-task impact, and provided evidence of functional scene awareness. ey96’s concerns are partially addressed.

Reviewer Wz4f’s major concern about novelty is not fully addressed.

For reviewer S7SK, the reviewer’s concern about limited evaluation beyond the constructed dataset is not fully addressed.

---

### Decision · Program_Chairs · 2026-01-26

Reject